# Generalized measurement error: Intrinsic and incidental measurement error

**Edward Kroc** [ID] *

Measurement, Evaluation, and Research Methodology, University of British Columbia, Vancouver, British Columbia, Canada

* ed.kroc@ubc.ca

## Abstract

In this paper, we generalize the notion of measurement error on deterministic sample data-sets to accommodate sample data that are random-variable-valued. This leads to the formulation of two distinct kinds of measurement error: *intrinsic* measurement error, and *incidental* measurement error. Incidental measurement error will be recognized as the traditional kind that arises from a set of deterministic sample measurements, and upon which the traditional measurement error modelling literature is based, while intrinsic measurement error reflects some subjective quality of either the measurement tool or the measurand itself. We define calibrating conditions that generalize common and classical types of measurement error models to this broader measurement domain, and explain how the notion of generalized Berkson error in particular mathematicizes what it means to be an *expert* assessor or rater for a measurement process. We then explore how classical point estimation, inference, and likelihood theory can be generalized to accommodate sample data composed of generic random-variable-valued measurements.

## Introduction

The purpose of this paper is not to introduce another measurement error model or specification, but to define a generalized concept of measurement error altogether. To accomplish this, we will separate the measurement and the sampling processes that are usually assumed to occur simultaneously in statistical situations, and we will allow for a *sample measurement process* to yield more general objects than real numbers. It is critical here to recognize that the traditional thinking in modern statistics is that one *samples* a certain (finite) number of elements (e.g. individuals, time points) from a fixed sample space (e.g. population, continuum of time), call these elements $\omega_1, \ldots, \omega_n$, and then observes the sample realizations of the target random variable of interest $X$ at these sample units: $X(\omega_1), \ldots, X(\omega_n)$. When we only observe some error-prone proxy $X^*$, then we instead record the sample realizations of $X^*$ at these same sample units: $X^*(\omega_1), \ldots, X^*(\omega_n)$. Seemingly all traditional and modern notions of measurement error follow from this *deterministic* observed data paradigm, and it is precisely this paradigm that this paper proposes to alter.

Specifically, we use the framework of *random-variable-valued measurements* (RVVMs) introduced in Kroc [1] to generalize what can be meant by measurement error. This is not a

**Data Availability Statement:** All relevant data were simulated using R; reproducible code appears within the submission's Supporting information files and a linked-to GitHub repository (https://github.com/edkroc/Supplemental_information_

S2_PLOSOne_
GeneralizedMeasurementError2023).

**Funding:** The author received no specific funding
for this work.

**Competing interests:** The author has declared no
competing interests exist.

statistical model, and this is not a proposed alternative to current methodologies; it is a proba-
bilistic framework that allows for a generalization of traditional measurement error modelling
to encompass novel sources of measurement uncertainty. In [1], Kroc's definition of this
framework was imprecise and relied on an untestable measurability assumption for most
applications; in this paper, a more precise and rigorous definition is proposed that does not
require this assumption. Details appear in the Methods section, but for now, it suffices to state
the following. Traditionally, whether one works with sample realizations of $X$ or some error-
prone proxy $X^*$, the resulting set of real numbers constitutes our *sample data*, and these sample
data are purely *deterministic* (i.e., they are fixed real numbers). All actual estimation, inference,
modelling, and prediction follow directly by plugging these real numbers into various formu-
lae for sample statistics or statistical models. What we relax in what follows is that we do not
require these sample data to form a set a real numbers. Instead, we will consider a generic *mea-
surement protocol* $\rho_X$ for a real-valued target (random) phenomenon of interest that is applied
to each of the sampled units $\omega_1, \ldots, \omega_n$. These sample measurements, $\rho_X(\omega_1), \ldots, \rho_X(\omega_n)$ are
allowed to be random variables. As was shown in Kroc [1], and as we will review in the next
section, such a formulation generalizes the classical sample data situation to allow for uncer-
tainty inherent to the *sample measurement process* itself; i.e., with this formulation, one may
encode measurement uncertainty at the level of the individual sample unit $\omega \in \Omega$, and such
measurement uncertainty is allowed to be potentially unique on each element. This capability
is desirable in a variety of applied data situations, and we discuss some of these in the subse-
quent text, though our main focus is on the mathematics of this generalized observed data
framework. The reader does not have to be familiar with the content of Kroc [1], although that
paper can be a useful reference for additional example applications of the RVVM framework.

Traditional and modern measurement error modelling all start from the same conceptual
principle: That one only observes some noisy proxy $X^*$ for a target random variable $X$ of actual
interest. Classically, these two random variables are often related via the simple algebraic rela-
tion

$$X^* = X + E, \tag{1}$$

where $X$ is the true, target quantity of inferential interest, $X^*$ is the actual, observable proxy for
$X$, and $E$ is the constituent error demanded by the additive expression. Up to exponentiation,
this algebraic relation captures the essence of multiplicative measurement error as well. The
random variables $X^*$, $X$, and $E$ are always assumed to live on the same (arbitrary) measurable
space $(\Omega, \mathcal{F})$, usually taken to be the reals imbued with the Borel $\sigma$-algebra, and are usually
assumed to be real-valued. Eq (1) is also usually understood to hold pointwise almost every-
where with respect to one of the probability measures attached to $X$ or $X^*$.

In applied modern practice, the algebraic relation in (1) is often replaced by a likelihood
specification, such as

$$f(x, x^*) = f(x^*|x)f(x), \ \text{or} \ f(x, x^*) = f(x|x^*)f(x^*). \tag{2}$$

Here, $f(x, x^*)$ is the joint density of $X$ and $X^*$ decomposed into the usual product of condi-
tional and marginal densities. The former decomposition is typically used when assuming clas-
sically calibrated measurement errors and the latter decomposition when assuming Berkson
calibrated errors [2]. Eq (1) is often implied by such a specification, though it need not neces-
sarily be so depending on the particular likelihoods one chooses to specify. This likelihood
specification has been expanded and adapted to host more complex statistical structures that
relate many random variables simultaneously, some subject to measurement error and some
perhaps not. In practice, one may be interested in building some kind of regression model for

a response $Y$ dependent on a predictor $X$ measured with measurement error and on other predictors $Z$ that are free of measurement error. Let $X^*$ be the noisy measurement for $X$. In this framework, one generally sees specifications like

$$f(y, x, z, x^*) = f(x^*|y, x, z)f(y|x, z)f(x|z)f(z), \qquad (3)$$

where the first factor is the conditional density of $X^*$ given $Y$, $X$, and $Z$, typically called the measurement model, the second factor is the conditional density of $Y$ given $X$ and $Z$, usually called the outcome or response model, and the third and fourth factors give the joint density of $X$ and $Z$, usually referred to as the exposure model. Most modern techniques in measurement error modelling can be expressed in such a generic form and enjoy a wide range of applications (see [2–5] for comprehensive treatments).

The generalized notion of measurement error defined in this paper leads to a novel and natural distinction between what can be thought of as *intrinsic* measurement error, and *incidental* measurement error. This latter variety of measurement error is precisely the classical kind that arises from the real-valued pointwise understanding of measurement error captured by Eqs (1), (2) or (3), while the former variety arises when the measurement error is not something that can necessarily be removed, even theoretically, from the measurement process by improving the quality of measurement in some way. Specifically, intrinsic measurement error allows one to encode measurement uncertainty at the level of a single sample unit within a sample space. Informally, one can imagine replacing the measurement model $f(x^*|y, x, z)$ of Eq (3) by one that looks like $f(x^*|\omega)$ for an arbitrary $\omega \in \Omega$. The probability action of the error-prone observable for $x$ can thus be unique to each sample unit in the sample space, rather than only unique to a subset of the sample space defined by a fibre of the conditioning variables $x$, $y$, and $z$. This has the effect of converting a modelling problem into a problem of data collection.

In the special case of a Bernoulli target phenomenon, the concept of intrinsic measurement error has already been long recognized and explored in the cognitive psychology literature, where the mathematical framework goes by the name of *confidence weighting* (e.g., see [6–9]). This literature allows one to build a simple theory of estimation for a Bernoulli target phenomenon when the measurement process is subject to intrinsic measurement error, but it does not allow for inference to a population (i.e., there is no treatment of the distribution or variance of a statistic), and it does not readily generalize to non-Bernoulli target phenomena, nor does it lend itself to coherent integration into a diverse model-building framework. As will be shown, however, the framework of RVVMs can tackle all these issues.

All treatments of measurement error invariably require additional data and/or a statistical model to relate the target quantity of interest $X$ to its observable and error-prone proxy $X^*$. For instance, in a validation study, the error-free values of $X$ are observed for a subsample. These can then be used to validate a proposed measurement error model relating $X$ to $X^*$ improving inferences dependent on $X$ when using only the observed values of $X^*$ on the entire sample [10, 11]. Replication studies require two or more error-prone proxies for $X$ which can then be combined to improve inferences dependent on $X$ via some kind of statistical model [12–14]. Notably, this paradigm is the one most commonly employed in factor analysis for the study of latent variables, ubiquitous in the scale development and survey literature of the behavioural sciences [15–17]. Calibration studies extend this idea to when some error-prone proxy is subject to systematic or differential measurement error, usually requiring a more complex regression framework to relate the various error-prone proxies to the true measurand of interest, perhaps dependent on other exogenous sources of information that could be measured with or without measurement error [18, 19].

The framework of RVVMs also requires additional data, but it does not necessarily require the specification of a statistical model relating the error-prone sample observations to the target measurand of interest. Specifically, to use RVVMs in a nontrivial way, we require additional information about the *confidence* or *certainty* that the sample measurement assumes a set of values. This additional information must be supplied by the measurement apparatus, as when a respondent answering a survey item tells us their subjective level of confidence in their response, say on a 0–100% scale. This extra "confidence information" is then combined with their point response (e.g., their chosen response to the survey item) to define a real-valued random variable. Typically, the point response may define this RVVM's mean and the confidence information may be used to define its variance. In some situations, this will require assuming a parametric distributional structure for the RVVM (so a statistical model), but in other situations the probabilistic structure will be automatically specified by the combination of sample data, in particular when the measurand is a categorical quantity. RVVMs thus require more information from the measurement apparatus to improve estimates and inferences, rather than necessarily requiring an analyst to propose a measurement error model.

After establishing the foundational definitions of the framework (Methods section), we will then expand on the work in [1] in the Results section by defining calibrating conditions that generalize the two most classical kinds of measurement error models, generally attributable to Pearson [20], Spearman [21], and Berkson [22]. We will discuss how the notion of generalized Berkson error in particular mathematicizes what it means to be an *expert* assessor or rater for a measurement process. This idea will be seen to have important practical consequences for data generation schemes involving any sort of trained expert assigning a static value to a random phenomenon, as when screening a patient for a particular affliction via a standardized questionnaire, recording morphological characteristics of an animal in an uncontrolled field environment, or validating psychological scales of latent, subjective phenomena. We will then explore how RVVMs can generalize the classical point estimation, inference, and likelihood theory, in particular the classical Bayesian and maximum likelihood approaches. This leads to the new notion of a generalized statistic that is a random variable defined over both a sampling space and a measuring space. We define the total variance of a generalized statistic, show how this generalizes the classical concept of standard error of a statistic, and define the general class of AA-estimators that enable a theory of point estimation over a sampling space for generalized statistics.

### Some motivating examples

It is likely worthwhile to spend some time describing some real world data generation schemes where RVVMs naturally arise. Here, I will discuss three examples: one in a health science context where the sample measurements are Bernoulli-valued, one in an educational context where the sample measurements are more general categorical-valued, and one in psychology where the sample measurements can be many of a variety of continuous random variables. See Kroc [1] for further examples in ecology and the social sciences.

**Movement skills in children via a clinically developed assessment tool.** The Test of Gross Motor Development, 3rd edition (TGMD-3) is a popular assessment tool applied to children to study their fundamental movement skills [23, 24]. A child is assessed on 13 fundamental movement skills (e.g., running, jumping, sliding) by a trained clinician who indicates whether or not the child exhibits a particular component of the skill correctly or not. For example, for the running skill, component 1 asks if the child's "Arms move in opposition to legs with elbows bent" and component 2 asks if there is a "Brief period where both feet are off the surface," in addition to two other components. The clinician's response to each TGMD-3

item is a simple binary: either the component is observed or it isn't. Two subscale scores and a total gross score are computed by summing all relevant component 0/1 scores after two trials of each component.

It is not difficult to imagine that instances commonly arise where a definitive yes/no response cannot be determined for a particular item. When such a situation occurs, the current recommendation is to have the child perform the motor skill again, but no amount of repetition can guarantee that a particular child will clearly, say, bend their elbows as their arms move in opposition to their legs or fail to do so. If one elbow is consistently bent while the other is not, does this count as a "success" or a "failure" to perform the task as prescribed? Consequently, it is desirable for clinicians to be able to assign a kind of *uncertainty score* to what they are observing.

Future work by the author and others examines this application in detail with real subjects and clinicians, but for our purposes, it suffices to recognize that what is desired here is a *Bernoulli-valued measurement protocol* rather than a simple binary 0/1 measurement protocol. Thus, rather than forcing clinical raters to always assign only a 0 or 1 to a particular component score, they instead may assign a *certainty of success* score (or a *confidence weight*) ranging from 0 to 1 over the reals. For subsequent analytical purposes, this score is then taken as the defining parameter of a Bernoulli random variable, one for each sample measurement. Note that this is *not* the same thing as simply redefining the TGMD component scale as an analogue 0 to 1 response. Such a response process would encode, for instance, that a sample subject performed 60% of a particular component correctly, whereas the RVVM encoding would capture that the clinician was 60% certain that the component was performed correctly. This may seem like a distinction without a difference, but as the contents of this paper will establish, the information contained in these two separate measurement protocols is quite different and naturally leads to different summary statistics and uncertainty quantifications.

**Calculus knowledge via a classroom examination.** A second natural application arises from the field of educational testing. Consider the common situation of giving students a classroom examination in multiple-choice format. For instance, we may ask a class of introductory calculus students the following:

$$Q : \text{If } \sin(xy) = x, \text{ then } \frac{dy}{dx} =$$

$$(a) \quad \frac{1}{\cos(xy)}$$

$$(b) \quad \frac{1}{x\cos(xy)}$$

$$(c) \quad \frac{1 - \cos(xy)}{\cos(xy)}$$

$$(d) \quad \frac{1 - y\cos(xy)}{x\cos(xy)}$$

$$(e) \quad \frac{y(1 - \cos(xy))}{x}$$

The classical multiple-choice response format would have each student identify one answer, and treat all students who identify the same answer as equivalent (i.e., *exchangeable*) for this particular item. However, it is easy to imagine different students identifying the same answer for very different reasons that have important bearing on how a teacher can assess the current state of their knowledge. For example, Student 1 may identify option (d) as correct with total

certainty, reflecting a true mastery of (introductory) implicit differentiation. Student 2 on the other hand may feel uncertain if they should apply the product rule in addition to the chain rule, so deduce that either (b) or (d) must be correct; arbitrarily, they select (d) as their answer. This correct answer now reflects only partial understanding of the mathematical operations involved, but the customary forced-response approach will not be able to encode this knowledge differential. Finally, consider that Student 3 may have not studied any of the course content on implicit differentiation, selecting the correct answer (d) only by uniformly-at-random chance.

An alternative measurement protocol would ask students to assign how *certain* they are that each of the 5 multiple choice options is the correct answer. Under this scheme, Student 1 would identify option (d) with 100% certainty and identify the remaining options as certainly false; whereas Student 2 may identify options (b) and (d) as correct with 50% certainty, but the other options as certainly false; while Student 3 could just assign a 20% certainty of correctness to each of the 5 response options. Formally, each student's response is now a categorical-valued measurement (on 5 categories), where the defining parameters of the random variable are supplied by the student's percentage responses (or confidence weights) for each option.

Notice that the RVVM encoding now allows one to easily distinguish the state of knowledge of the three student respondents, even though they would each supply the same answer, (d), if forced to provide only a single response. Moreover, each student's score on the item is now a better reflection of this state of knowledge, with Student 1 receiving maximum credit for their response, and Student 2 receiving some credit for correctly eliminating some of the incorrect responses. True, under this measurement scheme, Student 3 now receives some credit for a complete lack of knowledge, but it is interesting to note that they would receive *full credit* for this same lack of knowledge under the classical forced-response measurement scheme. On average, over many such items, the two measurement protocols would produce the same scores for such a student, as they would only identify correct answers 20% of the time via uniform-at-random forced guessing. The RVVM measurement scheme simply encodes this information more directly on *each item*, rather than relying on an averaging mechanism over many items to capture a total state of knowledge.

**Quantifying satisfaction with life via a questionnaire.**    For a final example, consider the clinically developed Satisfaction with Life Scale (SWLS) which has been claimed "to measure global cognitive judgments of one's life satisfaction" [25]. The scale consists of 5 items where participants indicate how much they agree or disagree with an item's statement in a 7-point Likert response format ranging from 1 = strongly disagree to 7 = strongly agree. Consider the third item of the SWLS: "I am satisfied with my life." Such a statement could be interpreted and contextualized within a multitude of ways depending on the particular sample respondent's reading, mood, and life experiences. Currently, a respondent simply assigns an integer score from 1 to 7 to indicate in some vague sense how much they agree with this statement. There is no ability to encode ambiguity or uncertainty of the response.

It should not be controversial that the notion of *agreement with a statement* is a continuous quantity. Indeed, one could easily adapt the response formats of the items of the SWLS (or any similar scale) to have 9, 10, or any integer $n > 1$ number of ordered categories, or simply impose an analogue (e.g., slider scale) response format. It is less obvious that this continuous phenomenon should occupy a compact space, but for simplicity (and to reflect previous thinking on analogue response formats) we suppose that agreement can be thought of as a continuous quantity on a compact interval, say [0, 1]. Thus, we now consider an alternative measurement protocol where respondents are asked to *draw a probability distribution* that reflects their feeling of agreement with the given item. One respondent could interpret the item as being totally unambiguous and so definitely declare that they are 100% satisfied with

their life, drawing a point-mass distribution concentrated at 1. Another respondent may feel quite satisfied with their life in some ways (e.g., relationships with family and friends) but very dissatisfied with life in other ways (e.g., work), drawing instead a bimodal distribution that could be approximated, say, by a Beta(0.5, 0.5) distribution. A third respondent may feel mostly ambivalent about their life satisfaction, drawing something very close to a normal curve with small variance centred at 0.5.

Of course, it may be no simple task to get respondents to draw reasonable probability distributions for responses, or indeed, to even understand what is being asked of them. However, this problem has been examined extensively in the statistical elicitation literature, where the elicitation of probability distributions to create informative priors has been desired (e.g., [26–28]). Regardless of practical complications, the theoretical motivation is all that we require for the present paper.

## Materials and methods

In this section we make the basic mathematical definitions that allow us to speak coherently of *generalized measurement error*, and derive some fundamental results that will be useful in our later applications. Let $X$ be a real-valued random variable defined on the (arbitrary) measurable space $(\Omega, \mathcal{F})$. Throughout, $X$ will be the target variable of interest.

**Definition 1**. *A **measurement protocol** $\rho_X$ for $X$ is a Borel-measurable measure-valued mapping such that*

$$\rho_X : \Omega \to \{\mu \in \mathcal{M}(\mathbb{R}, \mathcal{B}(\mathbb{R})) : |\mu| = 1, \ \mathrm{supp}(\mu) \subseteq \mathrm{EssRng}(X)\}.$$

That is, $\rho_X$ maps every element in the sample space $\Omega$ to a Borel probability measure on $\mathbb{R}$ whose support is contained in the image (i.e., the essential range) of $X$. This last condition ensures that the measurement actually reflects information about $X$. Technically, one can weaken this condition to include probability measures whose support up to a Borel-null set is contained in the range of $X$, but we will not need such flexibility.

One must clarify what is meant by requiring the measurement protocol to be Borel-measurable. This is clearly not with respect to the Borel sets on Euclidean space. Instead, we imbue the set of Borel probability measures on $\mathbb{R}$ with the topology generated by equipping the space with the total variation norm. Thus, one can consider the open balls in this metric:

$$B(\mu, r) := \left\{ v \in \mathcal{M}(\mathbb{R}, \mathcal{B}(\mathbb{R})) : |v| = 1, \ \sup_{B \in \mathcal{B}(\mathbb{R})} |\mu(B) - v(B)| < r \right\}.$$

The smallest topology containing all such open balls is the usual weak (or weak*) topology on the space of probability measures. And naturally, the smallest $\sigma$-algebra containing all such open balls is the Borel one on the space of probability measures. Restricting to measures with support contained in the always closed set of the essential range of $X$ yields the desired Borel $\sigma$-algebra, and it is with respect to this that we say the measurement protocol is Borel-measurable.

Since we can naturally identify any probability measure in the image of the measurement protocol with a real-valued random variable (distributed over $\mathbb{R}$ according to the measure), we call the elements of the image of the measurement protocol $\rho_X$ *random-variable-valued measurements* (RVVMs). When no confusion could arise, we may just write $\rho$ in place of $\rho_X$. In what follows, we will always use $\delta_x$ to denote the point-mass measure on $\mathbb{R}$ at $x \in \mathbb{R}$. Measurement protocols that generate only point-mass measures are said to generate *trivial* RVVMs; otherwise, the RVVMs are nontrivial. A measurement protocol is said to generate *mutually*

*independent measurements* if $\mu_\omega \perp \mu_\gamma$ for all $\omega \neq \gamma \in \Omega$. We will focus attention almost exclusively on mutually independent measurement protocols, though note that there are many real-world scenarios where such a simplification would not be reasonable (e.g. when observer and/or order effects can influence the measurement process).

We will often be interested in the fibres of the measurement protocol, and so define, for any given $\omega \in \Omega$, the set

$$\mathcal{O}_{\rho,\omega} = \rho^{-1}(\mu_\omega) = \{\omega' \in \Omega : \mu_{\omega'} = \mu_\omega\}.$$

When the measurement protocol is understood, we will just write $\mathcal{O}_\omega$. The following lemma will be very useful in what follows.

**Lemma 1**. *For any measurement protocol defined according to Definition 1, and any $\omega \in \Omega$, the fibre $\mathcal{O}_\omega$ is always $(\Omega, \mathcal{F})$-measurable.*

*Proof.* Fix $\omega \in \Omega$. We only need to show that the singleton $\{\mu_\omega\}$ is always a Borel set, since then the definition of a measurement protocol as a measurable mapping guarantees that its preimage lies in $\mathcal{F}$, the $\sigma$-algebra of the domain. The topology generated by the total variation norm gives us the open sets $B(\mu_\omega, r)$ for any $r > 0$, while the Borel $\sigma$-algebra generated by this topology guarantees the inclusion of countable intersections of these open balls. So since

$$\{\mu_\omega\} = \bigcap_{k=1}^{\infty} B(\mu_\omega, k^{-1}),$$

the singleton is also a Borel set. Thus, $\mathcal{O}_\omega \in \mathcal{F}$.

The following definition will be seen to generalize the traditional notion of measurement error.

**Definition 2**. *Fix a measurement protocol $\rho_X$ for X. If for (almost) every $\omega \in \Omega$ one has*

$$\rho_X(\omega) = \delta_{X(\omega)},$$

*then the measurement protocol $\rho_X$ is free of* **measurement error** (*almost everywhere with respect to the distribution of X*). *Otherwise, the measurement protocol is said to produce measurement error.*

It will be very important in practice to be able to coherently distinguish between two kinds of measurement error.

**Definition 3**. *Suppose $\rho$ is subject to measurement error, as defined in Definition 2. If $\rho$ generates only trivial RVVMs, then we say that $\rho$ generates only* **incidental measurement error**. *Otherwise, we say that $\rho$ generates* **intrinsic measurement error**.

Notice that trivial RVVMs can only encode *incidental* measurement error, by definition. RVVMs arising from such measurement protocols are the subject of traditional and modern measurement error modelling, but such techniques are not able to encode *intrinsic* measurement error.

Intrinsic measurement error is what Kroc [1] has previously referred to as *response process error*. However, I believe the new terminology is more general and appropriate. As the name suggests, intrinsic measurement error is something that is intrinsic to the measurement process itself, not simply an artifact of exogenous influence, random noise, or miscalibration. Often, such measurement error exists because of something intrinsic to the study phenomenon itself. For example, any measurement of an unobjectively defined phenomenon that relies on subjective judgement, like "biodiversity" or "anxiety," will contain intrinsic measurement error (whether or not an observer actually *chooses to encode* it or not).

**Example 1**. To best understand this, consider the simple goal of measuring an individual's height (in centimetres) using two different measurement protocols. For measurement protocol

$\rho_1$, an individual's height $X(\omega)$ is measured to the nearest centimetre using a rigid tape measure. This generates an observed measurement $X^*(\omega) \in \mathbb{R}$ subject to *incidental* measurement error. That is, $\rho_1(\omega) = \delta_{X^*(\omega)}$ is a point-mass measure, where $X(\omega)$ and $X^*(\omega)$ are real numbers that may differ due to rounding error and/or careless use of the rigid tape measure apparatus.

Now consider a second measurement protocol $\rho_2$ for an individual's height $X(\omega)$ where the person conducting the measurement simply eyeballs the target individual and assigns what they think is a plausible range of values in which $X(\omega)$ could lie. For example, if $X(\omega) = 180$, the person tasked with assigning the sample measurement may say that they are sure the person's height is somewhere in the range of 175 to 185 cm, with no subjective preference for any values in this range. Hence, $\rho_2(\omega) \sim Unif\,[175, 185]$. Clearly, this measurement contains *intrinsic* measurement error, and the relationship between the observed sample measurement and the true measurand *cannot* be expressed as in the classical pointwise expression of Eq (1), or by using the likelihood specifications of (2) or (3). In essence, we are "conditioning on the individual," $\omega$, *not* on the random variable, $X$ or $X^*$. Traditional notions of measurement error have always relied on some sort of implied exchangeability of sample individuals over common values of either the true measurand or its error-prone proxy (see, e.g., [29]). Generic measurement protocols, and so RVVMs, do not rely on any such exchangeability condition, which allows for greater ability to capture novel sources of uncertainty in a data collection process, particularly at the level of the individual sample unit.

For the particular example above, the intrinsic measurement error is an inherent feature of the measurement protocol $\rho_2$, not the target phenomenon of measurement itself. That is, an individual's height is an objectively defined quantity that is well defined and stable regardless of how one chooses to actually measure it (if at all); i.e., it is independent of the individual/ apparatus conducting the measurement. The intrinsic measurement error present in $\rho_2$ simply reflects the subjective nature of the measurement process defined by $\rho_2$ itself. Put another way, intrinsic measurement error is generated here only because the measurement process is suboptimal; the intrinsic measurement error could easily be removed by choosing a better measurement process/apparatus. However, there are many situations where the actual target of measurement lacks such objectivity in its definition, measuring "biodiversity" or "anxiety" for example. Such targets are usually quantified with a real number derived from some composite score over a set of items that attempt to measure (broadly construed) at least some part of the target quantity and/or its qualia.

**Example 2**. As a simple example, we could imagine measuring an individual's anxiety level at present by asking them to respond to the single item "How anxious are you feeling today?" on, say, a 0–100 analogue response scale, with 0 representing "not anxious at all" and 100 representing "maximally anxious." Customarily, one forces respondents to record only a single number for their response (which generates a trivial RVVM); however, such an approach suppresses the subjective uncertainty intrinsic to an individual respondent's answer to the item, uncertainty that reflects (1) the individual's subjective interpretation of the question being asked, the context of the item, and the metric structure of the proposed 0–100 scale, and (2) given such an interpretation, however stable or not, the individual's subjective feeling about their own level of anxiety. These sources of uncertainty are intrinsic to the measurand itself, and they cannot be removed by simply defining a "better" measurement tool (though perhaps their effects can be mitigated). The customary approach of forcing a single real-valued response for each respondent yields a trivial measurement protocol that structurally *ignores* these important and intrinsic sources of measurement uncertainty. A more informative measurement protocol would aim to record some of this subjective uncertainty in the sample measurement process, perhaps by allowing respondents to specify a range of plausible values for their present anxiety level and/or weight that range of values to define a subject-specific

probability measure (i.e., a nontrivial RVVM). Kroc [1] argues extensively for the practical benefits of this approach. Indeed, as we will see in the next section, it is often desirable to allow measurement protocols the ability to encode intrinsic measurement error, as it may often be easier to ensure a desirable *calibration* condition on the sample measurements, which can then yield unbiased inferences.

The following technical lemma will be useful subsequently.

**Lemma 2**. *Suppose ρ generates only trivial RVVMs. Then the Borel σ-algebra that ρ maps into has the discrete topology.*

*Proof*. We need to show that every subset of point-mass measures on $\mathbb{R}$ is open with respect to the total variation norm when the measurement protocol generates only trivial RVVMs. Since arbitrary unions of open sets are open in a topology, it suffices to show that the arbitrary singleton $\{\delta_x\}$ is open, for any fixed $x \in \mathbb{R}$. Consider the open ball:

$$B(\delta_x, r) := \left\{ \delta_y : \sup_{B \in \mathcal{B}(\mathbb{R})} |\delta_x(B) - \delta_y(B)| < r \right\}.$$

Since $(\mathbb{R}, \mathcal{B}(\mathbb{R}))$ is Hausdorff, there always exists a Borel set that separates $x, y \in \mathbb{R}$ for any $y \in \mathbb{R}$ when $y \neq x$. Consequently, we find that $B(\delta_x, r) = \{\delta_x\}$ for any $0 < r < 1$.

## Results

### Some useful calibrations of measurement protocols

The idea of *calibration* has a long history in the measurement and measurement error literature (e.g., see Chapter 4 of [5] or Chapters 1 and 7 of [2]), but the idea is simply that it is often desirable for an error-prone proxy and a measurand to be equated via some conditional expectation statement. Such a condition is usually required in applied practice to translate inferences about the error-prone proxy into inferences (often unbiased) about the target measurand of interest. Different calibration conditions are usually prerequisite for different measurement error models. The first calibration condition widely considered in the statistical literature can be attributed to Pearson [20] and Spearman [21], and requires that the random errors given by a random variable $E$ in Eq (1) must balance on all values of the target quantity of interest; i.e., they required [5]:

$$X^* = X + E, \quad \mathbb{E}(E \mid X) = 0. \tag{4}$$

Note that this condition is equivalent to requiring that $\mathbb{E}(X^* \mid X) = X$. In the literature, a measurement error model that assumes such a calibrating structure is referred to as *classical* (e.g. [2, 4]). Such a measurement error model is ideally suited to describe many kinds of measurement processes arising from studies of classical objects in the natural sciences: for example, when an observer records an astronomical observation of location via a telescope. In a very rough sense, the target quantity is fixed at whatever it is (here, perhaps the position of a planet relative to some star), and the astronomer makes a measurement of this target quantity (some distance) that can be compromised in various random and, in the frequentist sense, balanced ways: a slight miscalibration of the telescopic measuring apparatus, an imprecision induced by finite round-off error, a careless misreading of the measurement instruments by the astronomer, etc.—all error phenomena that we could reasonably expect to balance out to nothing more than added noise (i.e. variability) upon remeasurement. Variations on this error structure have also enjoyed wide application in the natural sciences (see [3]) as well as many social sciences like economics [30] and psychology [31], although often with additional structure assumed.

The other obvious conditional expectation one could require rather than (4) is the so-called Berkson measurement error structure [5]

$$X^* = X + E, \quad \mathbb{E}(E \mid X^*) = 0. \tag{5}$$

Again, note that this condition is equivalent to $\mathbb{E}(X \mid X^*) = X^*$. While indirectly considered earlier, Berkson [22] seems to have been the first to draw an explicit distinction between the kind of probability action in models (4) and (5). Berkson was primarily motivated to consider model (5) because it arises naturally in an analysis of *controlled experiments*. Indeed, in such a generic setup, a researcher typically fixes a certain number of set levels of an explanatory phenomenon and then records the output of a response variable over the different combinations of these set levels, often with replication, as in a factorial ANOVA design. For example, observing the temperature output of some mechanical apparatus under three different pressure conditions. The three different pressure conditions are set by the experimenter and are captured as three different values of $X^*$. But the true pressure of the apparatus, $X$, may vary slightly from this researcher-specified value for each replication due to imperfections in the controlling process, for example, ambient humidity and weather. Nowadays, such measurement error models are also natural to consider in many epidemiological contexts, for example when assigning a certain level of exposure of some contaminant to all individuals at a sampled location/site (see [2, 32] for more examples).

Now we define some *calibration* conditions that would be desirable for general measurement protocols to obey in order to make valid inferences about the target phenomenon of interest $X$ using only the observed measurements $\rho_X(\omega)$. These calibration conditions will be shown to generalize the Berkson and classical calibration conditions just described. Here and in what follows, we suppose $\omega \in \mathcal{S} \subset \Omega$ with $\#(\mathcal{S}) = n < \infty$, where $\mathcal{S}$ is generic notation for a (in practice, finite) sample of the overall sample space $\Omega$.

**Berkson calibration.** **Definition 4**. *Fix some $q \geq 1$. We say that a measurement protocol is* **qth order Berkson calibrated** *to $X$ if for every $\omega \in \Omega$, the qth moment of the measure $\rho_X(\omega) = \mu_\omega$ is a version of the conditional expectation of $X^q$ over $\mathcal{O}_\omega$; i.e.,*

$$\text{for all } \omega \in \Omega, \quad \mathbb{E}(X^q \mid \mathcal{O}_\omega) = \int_{\mathbb{R}} x^q \, d\mu_\omega(x).$$

Note that the conditional expectation in this definition is well-defined by Lemma 1. The reason for the "Berkson" part of this terminology is due to the following.

**Proposition 1**. *First order Berkson calibration generalizes the second type of classical measurement error model described in* Eq (5), *originally formalized by Berkson* [22]. *In particular, the real-valued random variable $X^*$ is classically Berkson calibrated to $X$ if and only if $X^*$ is generated by a trivial measurement protocol that is first order Berkson calibrated.*

*Proof.* All equations that follow should be understood to hold pointwise almost everywhere. As in the preamble to this section, the classical Berkson measurement error model can be defined as $\mathbb{E}(X \mid X^*) = X^*$.

First suppose that we have a real-valued $X^*$ (Borel measurable) that is classically Berkson calibrated to $X$; i.e., suppose that $\mathbb{E}(X \mid X^*) = X^*$ holds. Since $X^*$ is a well-defined observable proxy for $X$ in the classical sense (i.e., free of intrinsic measurement error), we can define a measurement protocol $\rho$ for $X$ via $\rho(\omega) := \delta_{X^*(\omega)}$ for every $\omega \in \Omega$. This is a trivial measurement protocol. To see that it is first order Berkson calibrated, just note that

$$\mathcal{O}_\omega = \rho^{-1}(\delta_{X^*(\omega)}) = (X^*)^{-1}[X^*(\omega)],$$

which is automatically $(\Omega, \mathcal{F})$-measurable, since $X^*$ is Borel measurable. So now, using the

fact that $X^*$ is classically Berkson calibrated to $X$, we have

$$\mathbb{E}(X \mid \mathcal{O}_\omega) = \mathbb{E}(X \mid X^*)(\omega) = X^*(\omega) = \int_\mathbb{R} x \; d\delta_{X^*(\omega)},$$

so $\rho$ is first order Berkson calibrated.

The converse follows along the same lines with one extra step. Suppose we start with a trivial measurement protocol $\rho$ for $X$ that is first order Berkson calibrated. Since $\rho(\omega)$ is always a point-mass measure, for every $\omega \in \Omega$, there is a unique $r_\omega \in \mathbb{R}$ such that $\rho(\omega) = \delta_{r_\omega}$. So we may define the new random variable $X^*$ on the measurable space $(\Omega, \mathcal{F})$ via $X^*(\omega) = r_\omega$. Moreover, this random variable is guaranteed to be measurable by Lemma 2. To see why, take any Borel set $B$ in $\mathbb{R}$. Notice:

$$\begin{aligned}
(X^*)^{-1}(B) &= \{\omega \in \Omega : X^*(\omega) \in B\} \\
&= \bigcup_{b \in B}\{\omega \in \Omega : \rho(\omega) = \delta_b\} \\
&= \bigcup_{b \in B}\rho^{-1}(\delta_b) \\
&= \rho^{-1}\left(\bigcup_{b \in B}\{\delta_b\}\right)
\end{aligned}$$

By Lemma 2, the set $\bigcup_{b \in B}\{\delta_b\}$ is open, and so is also Borel. Thus, by Definition 1 its inverse image is $(\Omega, \mathcal{F})$-measurable.

Now that $X^*$ is an honest random variable on $(\Omega, \mathcal{F})$, we have

$$X^*(\omega) = r_\omega = \int_\mathbb{R} x \; d\delta_{r_\omega} = \mathbb{E}(X \mid \mathcal{O}_\omega),$$

since we have assumed first order Berkson calibration. But by the same argument as before, we have

$$\mathbb{E}(X \mid \mathcal{O}_\omega) = \mathbb{E}(X \mid (X^*)^{-1}(r_\omega)) = \mathbb{E}(X \mid X^*)(\omega).$$

Thus, $\mathbb{E}(X \mid X^*) = X^*$ and $X^*$ is classically Berkson calibrated.

For the important special case of Bernoulli-valued measurements, there is only one kind of Berkson calibration, as such a measurement protocol is first order Berkson calibrated if and only if it is $q$th order Berkson calibrated for any other $q > 1$. First order Berkson calibration is dealt with extensively in Kroc [1], and mathematically encodes the idea of a measurement protocol being generated by an *expert observer/assessor* for Bernoulli-valued measurements. Notably, this is also the kind of calibration considered in the confidence weighting literature of cognitive psychology (e.g., see [7, 8]).

It is interesting to note that this interpretation aligns well with the usual interpretation of classical Berkson measurements; i.e., assigning the expected value of the group to each individual member of the group, while the true (unobserved) value of the individual can vary with zero expectation about the observed group value. But rather than the typical application, as when assigning the same level of radon exposure at a test site to all individuals who live there (a typical epidemiological application, see [32]), we have in mind an expert assessor who, say, assigns the same probability of sex to all members of a monomorphic species that fit the same phenotypic criteria (see the extended example in Section 4 of Kroc [1]). The hypothetical population of individuals fitting that criteria is the "group," and each individual gets assigned the

same expected value. For this sex example, that value is some $\theta \in [0, 1]$, and first order Berkson calibration holds if precisely $\theta$ of those individuals in the group are actually the target sex.

Such Bernoulli-valued measurements are very common in ecology and psychology, but of course none of the above framework is that restrictive. Thus, what we mean by "expert" can be much more varied for general measurement protocols. In particular, as we will see with an example in the next section, it may often be most reasonable to define an "expert assessor" in general as one who generates a measurement protocol that is $q$th order Berkson calibrated *for all $q \geq 1$*.

**Classical calibration.**   Define the set

$$\mathcal{L}_\omega = \{\omega' \in \Omega : X(\omega') = X(\omega)\}.$$

Note that one could write $\mathcal{L}_\omega = X^{-1}[X(\omega)]$, and so $\mathcal{L}_\omega$ is automatically $(\Omega, \mathcal{F})$-measurable. Notice that the $\sigma$-algebra generated by this collection of sets is actually equal to the $\sigma$-algebra generated by the target quantity $X$; i.e., $\sigma(\mathcal{L}_\omega) = \sigma(X)$. Note also that $\mathcal{L}_\omega$ does not depend on the choice of measurement protocol $\rho_X$.

**Definition 5**. *Fix some $q \geq 1$. We say that a measurement protocol is* **$q$th order classically calibrated** *to $X$ if for every $\omega \in \Omega$:*

$$\mathbb{E}\left[\int_\mathbb{R} x^q \, d\mu_\gamma(x) \ \Big| \ \mathcal{L}_\omega\right] = X^q(\omega).$$

**Proposition 2**. *First order classical calibration generalizes the first type of classical measurement error model described in* Eq (4), *attributable to Pearson* [20] *and Spearman* [21]. *In particular, the real-valued random variable $X^*$ is classically calibrated to $X$ if and only if $X^*$ is generated by a trivial measurement protocol that is first order classically calibrated.*

*Proof.* Again, all equations that follow hold pointwise almost everywhere. As in the preamble to this section, the classical measurement error model holds if $\mathbb{E}(X^* \mid X) = X$.

First suppose that we have a real-valued $X^*$ (Borel measurable) that is classically calibrated to $X$. As before, we can then define a trivial measurement protocol via $\rho(\omega) := \delta_{X^*(\omega)}$ for every $\omega \in \Omega$. Since one can write $\mathcal{L}_\omega = X^{-1}[X(\omega)]$, we have

$$\mathbb{E}\left[\int_\mathbb{R} x \, d\mu_\gamma(x) \ \Big| \ \mathcal{L}_\omega\right] = \mathbb{E}\left[\int_\mathbb{R} x \, d\delta_{X^*(\gamma)}(x) \ \Big| \ X^{-1}[X(\omega)]\right]$$
$$= \mathbb{E}(X^* \mid X)(\omega) = X(\omega),$$

so the measurement protocol is first order classically calibrated.

The converse also follows as in the proof of Proposition 1. Starting with a trivial measurement protocol $\rho$ for $X$ that is first order classically calibrated, for every $\omega \in \Omega$, we can always find a unique $r_\omega \in \mathbb{R}$ such that $\rho(\omega) = \delta_{r_\omega}$, and thus define the new random variable $X^*$ on $(\Omega, \mathcal{F})$ via $X^*(\omega) = r_\omega$. Note that this random variable is well-defined and $(\Omega, \mathcal{F})$-measurable by the same application of Lemma 2 as in the proof of Proposition 1. Then $X^*$ is classically calibrated by the same argument as above, rearranged:

$$X(\omega) = \mathbb{E}\left[\int_\mathbb{R} x \, d\mu_\gamma(x) \ \Big| \ \mathcal{L}_\omega\right]$$
$$= \mathbb{E}\left[\int_\mathbb{R} x \, d\delta_{X^*(\gamma)}(x) \ \Big| \ X^{-1}[X(\omega)]\right]$$
$$= \mathbb{E}(X^* \mid X)(\omega).$$

**Proposition 3**. *Let ρ be a trivial measurement protocol for X. Then if ρ is free of measurement error (i.e. free of incidental measurement error), then ρ is both qth order Berkson calibrated and qth order classically calibrated for all q ≥ 1. The converse also holds.*

This is a slight generalization of a classic result in the study of measurement error—i.e., there is no (incidental) measurement error if and only if an observable proxy is simultaneously Berkson and classically calibrated—now rephrased with the general language of measurement protocols. This classical result is actually somewhat counterintuitive from a finite sample point of view, when one considers graphing a sequence of sample measurements of *X* and its proxy *X\** as sample realizations that fall "randomly" around the diagonal *x* = *x\** (see Fig 1). From a sample point of view, it seems that such a measurement is both Berkson and classically calibrated, but in fact this is not true. The apparent contradiction is resolved by realizing that such a picture describes a *sample* situation, not a population-level one. And even at that sample level, near the boundaries of the scatterplot, at most one of the Berkson or the classical Pearson/Spearman-type conditional expectation conditions can hold.

*Proof of Proposition 3*. We prove the forward direction first. Notice that $\mathcal{O}_{\omega} = X^{-1}[X(\omega)] = \mathcal{L}_{\omega}$, simply because we have required our measurement protocol ρ to always return the appropriate fixed value of *X* upon measurement. But now

$$\mathbb{E}(X^q \mid \mathcal{O}_{\omega}) = \mathbb{E}(X^q \mid X^{-1}[X(\omega)]) = X^q(\omega) = \int_{\mathbb{R}} x^q \, d\delta_{X(\omega)}(x),$$

for any ω, since the random variable *X* is always constant on the set $X^{-1}[X(\omega)]$. The

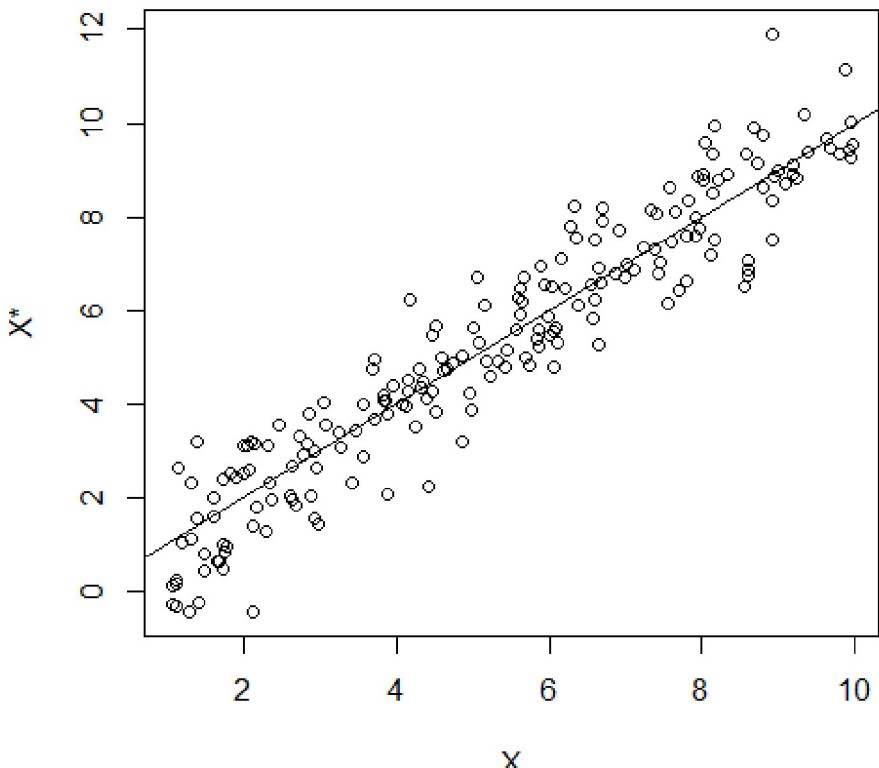

**Fig 1. An example of trivial RVVMs that appear, at first glance, to be simultaneously Berkson and classically calibrated, but can only actually be at most classically calibrated.**

measurement protocol is thus $q$th order Berkson calibrated. Moreover, $\int_{\mathbb{R}} x^q \, d\delta_{X(\gamma)}(x)$ is constant when $\gamma \in \mathcal{L}_\omega$, and in fact $X(\gamma) = X(\omega)$ for any such $\gamma$; thus, the measurement protocol is also $q$th order classically calibrated.

To prove the converse, note that $\sigma(\mathcal{L}_\omega : \omega \in \Omega) = \sigma(X)$ in general. Also, since $\rho$ generates only trivial RVVMs, we can introduce the notation $\rho(\omega) = \delta_{X^*(\omega)}$ for some random variable $X^*$ defined on $(\Omega, \mathcal{F})$. Note again that this is a well-defined $(\Omega, \mathcal{F})$-measurable random variable by Lemma 2 as in the proofs of Propositions 1 and 2. Using Propositions 1 and 2, we know that both $X^* = \mathbb{E}(X \mid X^*)$ and $X = \mathbb{E}(X^* \mid X)$. The first of these equalities implies that $\sigma(X^*) \subseteq \sigma(X)$, while the second equality implies $\sigma(X) \subseteq \sigma(X^*)$. So $\sigma(X) = \sigma(X^*)$ and using either of the previous two equations we find that $X = X^*$ a.e.

Notice that this proof also establishes that if $\rho$ is a trivial measurement protocol that is $q$th order Berkson/classically calibrated for some $q \geq 1$, then it is also $p$th order Berkson/classically calibrated for all $p \geq 1$. Notice also that one cannot remove the condition that the measurement protocol generates only trivial RVVMs in the statement of Proposition 3; that is, if a generic measurement protocol is both $q$th order Berkson calibrated and $p$th order classically calibrated for some $q, p \geq 1$, then this is *not* sufficient to imply that the measurement protocol must actually be free of measurement error. As an easy counterexample, suppose $X \sim N(0, 1)$ and $\rho(\omega) \sim N(X(\omega), \sigma_\omega)$. Then this measurement protocol is both first order Berkson calibrated and first order classically calibrated, but is obviously not free of (intrinsic) measurement error.

## Implications for estimation and inference

In this section, we take up the task of determining how one can actually use general RVVMs to perform estimation and inference generically. The first two subsections focus on the important special cases of generalized estimators for the mean and variance of a target phenomenon. The third subsection introduces the concept of a generalized statistic, its corresponding point estimator (referred to as an *AA-estimator*), and the total variance of a statistic that incorporates both uncertainty due to sampling error (*sampling variance*) and uncertainty due to measurement error (*measuring variance*). The final subsection aims only to introduce terminology and concepts for a likelihood-based theory for statistics derived from sample RVVMs.

**Estimating the mean of X.** Consider how we would define the RVVM generalization of the classical (unbiased) sample mean estimator. Kroc [1] shows that, assuming only first order Berkson calibration, the sample estimator

$$\overline{X}_\rho := \frac{1}{n} \sum_{i=1}^n \int_{\mathbb{R}} x \, d\mu_{\omega_i}(x) \tag{6}$$

is a natural generalization of the traditional sample mean $\overline{X} := \frac{1}{n} \sum_{i=1}^n X(\omega_i)$ for general RVVMs; i.e., $\overline{X}_\rho$ is an unbiased and consistent estimator of $\mathbb{E}(X)$ (so obeys a weak law of large numbers), and reduces to $\overline{X}$ if $\rho$ generates only trivial RVVMs, as long as $\rho$ is first order Berkson calibrated. This is easily seen by observing that first order Berkson calibration means that one can rewrite the sample mean estimator as

$$\overline{X}_\rho = \frac{1}{n} \sum_{i=1}^n \mathbb{E}(X \mid \mathcal{O}_{\omega_i}).$$

Taking an expectation immediately shows that the estimator is unbiased, and using the

standard expression for total variance (and recalling that we are assuming all measurement protocols generate mutually independent measurements), we have

$$\mathrm{Var}(\overline{X}_\rho) = \frac{\mathrm{Var}(X)}{n} - \frac{1}{n^2}\sum_{i=1}^{n}\mathbb{E}[\mathrm{Var}(X \mid \mathcal{O}_{\omega_i})], \tag{7}$$

which goes to zero as long as $\mathrm{Var}(X) < \infty$ since we always have $0 \le \mathbb{E}[\mathrm{Var}(X \mid \mathcal{O}_\omega)] \le \mathrm{Var}(X)$.

Notice also that a Central Limit Theorem applies to $\overline{X}_\rho$ under first order Berkson calibration if we assume some kind of regularity condition on the variability of the RVVMs. One could impose a Lyapunov or Lindeberg-type condition, but for simplicity just assume $\mathrm{Var}(X) < \infty$. Then note that $\overline{X}_\rho$ is an arithmetic average of the random variables $\mathbb{E}(X \mid \mathcal{O}_{\omega_k})$, which we can call $W_k$. Then $\mathbb{E}(W_k) = \mathbb{E}(X) < \infty$, and $\mathrm{Var}(W_k) = \mathrm{Var}(X) - \mathbb{E}[\mathrm{Var}(X \mid \mathcal{O}_{\omega_k})] \le \mathrm{Var}(X) < \infty$ for all $k$, assuming $X$ has finite first and second moments.

Berkson calibration, or indeed any kind of calibration condition that requires only that the RVVMs are "correct on average" in some sense, is not going to be sufficient to establish *consistency* (or asymptotic normality) of more general estimators to estimands. For instance, the WLLN is sufficient to establish consistency of classical MLEs (under the usual, mild regularity conditions), but the equivalent for RVVMs will *not* suffice. This is because in the classical, deterministic sample data setting, all observed variability comes from *sampling variability*, i.e., from the fact that different observations can be different numbers (the same is true with classical measurement error, with just an extra step establishing structure between the observed $X^*$ and unobservable $X$). But for RVVMs, each sample observation has a potential amount of intrinsic measurement error attached, and there is no way simple arithmetic combinations of these measures will be necessarily enough to "wipe away" this extra uncertainty / probability mass, at least not without assuming some kind of stabilizing structure to the RVVMs. We will investigate some of this new structure in the subsequent pages.

Nevertheless, if we want, we can construct approximate confidence intervals using the Central Limit Theorem for large enough sample sizes in the usual way using the expression just derived for the standard error of the generalized sample mean, Eq (7). For measurement protocols that are first order Berkson calibrated, this is more easily expressed without using the expression for total variance, resulting in the generic estimator

$$\widehat{\mathrm{Var}}(\overline{X}_\rho) = \frac{1}{n^2}\sum_{i=1}^{n}\widehat{\mathrm{Var}}[\mathbb{E}(X \mid \mathcal{O}_{\omega_i})] = \frac{\widehat{\mathrm{Var}}[\mathbb{E}(X \mid \mathcal{O}_\omega)]}{n}.$$

Absent calibration, using (6) the estimated standard error is simply

$$\widehat{\mathrm{Var}}(\overline{X}_\rho) = \frac{1}{n^2}\sum_{i=1}^{n}\widehat{\mathrm{Var}}\left[\int_{\mathbb{R}} x\, d\mu_{\omega_i}(x)\right] = \frac{1}{n}\widehat{\mathrm{Var}}\left[\int_{\mathbb{R}} x\, d\mu_\omega(x)\right].$$

The obvious unbiased sample estimator here is:

$$\widehat{SE}^2(\overline{X}_\rho) := \frac{1}{n(n-1)}\sum_{i=1}^{n}\left[\int_{\mathbb{R}} x\, d\mu_{\omega_i}(x) - \overline{X}_\rho\right]^2$$

This is an unbiased estimator for $SE^2(\overline{X}_\rho)$ under first order Berkson calibration. As we shall shortly see though, the sample standard error of this generalized mean estimator is *not* equal to just some sample size-scaled multiple of the generalized sample variance. Importantly, realize that calibrated inference on the mean of $X$ only requires *first order* Berkson calibration; that

condition is strong enough to imply unbiasedness of the generalized sample mean $\overline{X}_\rho$ and unbiasedness of the natural estimator of its standard error. This combined with the fact that a CLT applies means that all the classical inferential apparatus works as it should for the mean of $X$. In particular, we do not require control over the higher order properties of the sample measurements $\mu_\omega$ to estimate the standard error (and so construct approximate confidence intervals, etc.). This is a bit counterintuitive at first, since we are so used to thinking about the standard error of the sample mean as a function of the variance of $X$, but we will see that this intuition is in fact specious, an artifact of the triviality of the classical measurement protocol employed when working with classical deterministic sample data.

**Example 3**. We pause for a simple example that we will return to in the subsequent sections to illustrate new definitions and concepts. Let $X$ be Bernoulli so that any measurement protocol $\rho$ can only generate Bernoulli-valued measurements (some, of course, could be degenerate and produce trivial RVVMs). Consider the measurement protocol of total ambivalence; i.e., suppose $\rho(\omega) \sim Ber(1/2)$ for all $\omega \in \Omega$. Then $\int_\mathbb{R} x \; d\mu_\omega(x) = 0.5$ for all $\omega$, and so $\overline{X}_\rho = 0.5$ as well for any sample $\mathcal{S} \subset \Omega$. Thus, its corresponding standard error (both theoretical and estimated) is identically zero. This of course makes perfect sense since such a measurement protocol generates the exact same (rather useless) sample data for every single sample data set regardless of sample size. The generalized sample mean is a constant, so there is no variation in its value over different samples.

We make two additional remarks. First, notice that, absent Berkson calibration, the standard error of the mean need not shrink as sample size increases. In general, in order to guarantee such behaviour, one would need to impose some kind of regularity condition on the sequence of expectations of observed RVVMs; something like

$$\text{Var}\left[\int_\mathbb{R} x \; d\mu_\omega(x)\right] = o(n).$$

Second, we note that of course the proposed generalized sample mean $\bar{X}_\rho$ may not be the *only* generalized estimator one could consider. For *non*-independent measurement protocols, there are in fact (at least) two incomparable natural estimators one would want to consider. But for independent measurement protocols, the two natural choices actually reduce to the same estimator. Recall that we defined

$$\overline{X}_\rho := \frac{1}{n}\sum_{i=1}^n \int_\mathbb{R} x \; d\mu_i(x),$$

which is an empirical expectation with respect to all observed *marginal* (sample) distributions.

One could also consider the estimator

$$\text{AA}(\overline{X}_\rho) := \int_{\mathbb{R}^n} f(\mathbf{x}) \; d\mu_{\otimes_n}(\mathbf{x}), \tag{8}$$

for some function $f : \mathbb{R}^n \to \mathbb{R}$, where $\mu_{\otimes_n}$ denotes the joint probability measure, $\mu_1 \otimes \mu_2 \otimes \cdots \otimes \mu_n$ (the reason for the "AA" notation will be clarified in the subsection entitled "A natural method for generalized statistics with RVVMs"). This is now an estimator that depends on the *joint* distribution of the observed (sample) data, reflected in the joint structure of the measure $\mu_{\otimes_n}$. For estimating the mean of $X$, the natural choice of integrand is $f(\mathbf{x}) = \frac{1}{n}\sum_{i=1}^n x_i$, so by

linearity, one has

$$\int_{\mathbb{R}^n} \frac{x_1 + \cdots + x_n}{n} \; d\mu_{\otimes_n}(\mathbf{x}) = \frac{1}{n} \int_{\mathbb{R}^n} \sum_{i=1}^n x_i \; d\mu_1(x_1) \cdots d\mu_n(x_n)$$
$$= \frac{1}{n} \sum_{i=1}^n \int_{\mathbb{R}} x \; d\mu_i(x),$$

since independence of the measurement protocol ensures that the information contained in the joint distribution $\mu_{*_n}$ is *equivalent* to the information contained in the set of marginal distributions $\{\mu_i : 1 \le i \le n\}$, a fact reflected in the corollary that $\text{AA}(\overline{X}_\rho) = \overline{X}_\rho$. Once again note though that these two estimators will not necessarily reduce to the same expression if one considers non-independent measurement protocols.

**Example 4**. For instance, consider a Bernoulli-valued measurement protocol where subsequent sample measurements are subject to a form of *recency bias*. Such a situation could arise, say, if an insufficiently skilled assessor is asked to record the sex (F/M) of a sequence of captured monomorphic birds in a bird-banding field operation. Such an assessor may have a tendency to more likely assign a value of "F" to an individual bird immediately after assigning a value of "M" to a previous individual. For a sequence of $n$ sample measurements on $n$ distinct sample individuals, $\omega_1, \ldots, \omega_n$, the recorded RVVMs could be distributed according to the following rule:

$$\mu_1 \sim Ber(p_1), \quad \mu_{j|(j-1)} \sim Ber(q_i), \quad \text{where} \quad q_j = p_j - (p_{j-1} - 0.5)p_j(1 - p_j),$$

for $1 < j \le n$, and for a given sequence of parameters $p_1, \ldots, p_n \in [0, 1]$ that capture the assessor's judgment of probability that the sample individual is female if that particular individual was the first one to be measured (so, no recency bias could affect the sample measurement process). Now, the joint measure $\mu_{\otimes_n}$ can be decomposed into a product of conditional measures so that

$$\text{AA}(\overline{X}) = \frac{1}{n} \int_{\mathbb{R}^n} \sum_{i=1}^n x_i \; d\mu_{n|(n-1)}(x_n|x_{n-1}) \cdots \mu_{2|1}(x_2|x_1)\mu_1(x_1) = \frac{1}{n} \sum_{i=1}^n q_i,$$

while in order to compute $\overline{X}$, we would first need to derive the distributions of the marginal measures $\mu_i$, which will not equal the above conditional measures by virtue of the dependence structure induced by the recency bias of the assessor.

**Estimating the variance of X.** Now consider the problem of estimating $\text{Var}(X)$ using the RVVMs generated from a generic measurement protocol. As we will shortly see, the natural generalization of the traditional (unbiased) sample variance estimator

$$s^2 := \frac{1}{n-1} \sum_{i=1}^n \left(X(\omega_i) - \overline{X}\right)^2$$

is only unbiased if one assumes both first order *and* second order Berkson calibration simultaneously. To see why, first write

$$\text{Var}(X) = \text{Var}[\mathbb{E}(X \mid \mathcal{O}_\omega)] + \mathbb{E}[\text{Var}(X \mid \mathcal{O}_\omega)],$$

and then notice that first order Berkson calibration says

$$\mathbb{E}(X \mid \mathcal{O}_\omega) = \int_{\mathbb{R}} x \; d\mu_\omega(x),$$

while adding second order Berkson calibration yields

$$\text{Var}(X \mid \mathcal{O}_\omega) = \int_\mathbb{R} x^2 \, d\mu_\omega(x) - \left( \int_\mathbb{R} x \, d\mu_\omega(x) \right)^2.$$

Thus, one would consider

$$S_\rho^2 := \frac{1}{n-1} \sum_{i=1}^n \left[ \int_\mathbb{R} x \, d\mu_{\omega_i}(x) - \overline{X}_\rho \right]^2 + \frac{1}{n} \sum_{i=1}^n \left[ \int_\mathbb{R} x^2 \, d\mu_{\omega_i}(x) - \left( \int_\mathbb{R} x \, d\mu_{\omega_i}(x) \right)^2 \right] \quad (9)$$

as the natural generalization of the classical (unbiased) sample variance estimator. Notice that the first term here captures what we could naively characterize as the sample estimate of the true variance $\text{Var}(X)$ captured by the incidental measurement error in the RVVMs, while the second term captures the additional variance/uncertainty due to the less than total information present in the RVVMs; i.e., due to the intrinsic measurement error. In the traditional case of classical or Berkson measurement error on trivial RVVMs, the second term disappears and the first term captures both "sampling error" and "measurement error," confounded. Absent all measurement error, the second term again disappears and the first term captures simply "sampling error."

One can take an expectation to verify that $S_\rho^2$ is actually an unbiased estimator for $\text{Var}(X)$ for general RVVMs. This need not hold though unless we assume both first order and second order Berkson calibration; without second order Berkson calibration, we do not have average control over the second term in the expression of the estimator $S_\rho^2$, unless in the particular (but important) special case when all nontrivial RVVMs are Bernoulli-valued.

So, if we demand an "expert assessor" not only generate sample measurements that are accurate on average (according to the weak law of large numbers), but also for those sample units where a lot of measurement uncertainty is present, we require that the true values of the phenomenon $X$ are actually quite variable, then we need both orders of calibration. This type of control would certainly be desirable in model building/comparison contexts since then one cares about accurately capturing the structure of a (model) residual error.

**Example 5**. As a simple example of the general idea that second order Berkson calibration captures something meaningful about what we would expect "expert" assessments to look like (outside of the Bernoulli-valued context), consider the simplest case where the issue isn't degenerate: a measurement protocol that generates trinomial-valued measurements. For concreteness, say the phenomenon of interest is the species of gull observed at a survey site outside Vancouver, Canada. Due to observer distance from the sampled bird and known morphological ambiguities (present especially in winter months), even expert assessors can find it impossible to distinguish definitively between gulls of the Glaucous-winged, Herring, and Thayer's species [33], encoded as -1, 0, and 1 respectively.

Consider the fibre $\rho^{-1}(Cat_{(1/3,1/3,1/3)})$, containing all birds where the expert assessor is totally ambivalent about species classification; here, we use the $Cat_{(p_{-1},p_0,p_1)}$ notation to denote a categorical measure on three atoms. For simplicity, suppose there are only 6 birds total in this fibre, $\{\omega_i : 1 \le i \le 6\}$, with $X(\omega_i) = 0$ for $1 \le i \le 4$, $X(\omega_5) = -1$, and $X(\omega_6) = 1$. Thus,

$$\int_\mathbb{R} x \, d\mu_\omega(x) = 0 = \mathbb{E}(X \mid \mathcal{O}_\omega)$$

for all $\omega$ in $\mathcal{O}_\omega$. But

$$\int_\mathbb{R} x^2 \, d\mu_\omega(x) = \frac{2}{3} \ne \mathbb{E}(X^2 \mid \mathcal{O}_\omega) = \frac{1}{3}.$$

Indeed, this measurement protocol seems to be less than ideally "expert," since we would hope that part of "expert" means that on the fibre $\rho^{-1}(Cat_{(1/3,1/3,1/3)})$ the three species should be uniformly distributed. This isn't the case here. If we enlisted perhaps a "better trained" expert assessor, then they might identify these 6 birds with the sample measurement $Cat_{(1/6,2/3,1/6)}$, and then first and second order Berkson calibration would simultaneously hold.

**Example 6**. A second example will illustrate how estimating the standard error of the mean of $X$ only indirectly depends on estimating the variance of $X$, contrary to classical intuition. Consider again the example of a Bernoulli target phenomenon $X$ and the measurement protocol of total ambivalence; i.e., $\rho(\omega) \sim Ber(1/2)$ for all $\omega \in \Omega$. As noted in the previous section, one has $\overline{X}_\rho = 0.5$ for any sample $\mathcal{S} \subset \Omega$ of any size. Thus, the estimator's corresponding standard error (both theoretical and estimated) is identically zero. Nevertheless, any sample drawn under this measurement protocol certainly contains plenty of evidence of variation in the target phenomenon $X$, and we would estimate

$$S_\rho^2 = 0 + \frac{1}{n}\sum_{i=1}^n [0.5 - (0.5)^2] = 0.25.$$

Note that these calculations do not assume any kind of Berkson (or any other kind of) calibration.

**A natural method for generalized statistics with RVVMs.** The definitions of the estimators considered in the previous two subsections were motivated by a desire for unbiasedness under some type of Berkson calibration. However, there is a natural way to redefine any point estimator—and more generally, any statistic—to accommodate nontrivial RVVMs regardless of calibration considerations.

For a random sample $\mathcal{S} \subset \Omega$, let $\varphi(\mathcal{S})$ be a statistic, which in this very generic context simply means a function (not necessarily real-valued). Since $\mathcal{S}$ is random, $\varphi(\mathcal{S})$ is also a random variable. In the traditional fixed sample data context (i.e., measurement protocols that generate only trivial RVVMs), one then observes a particular sample $\mathcal{S}_{obs}$ of real numbers, and then the sample statistic $\varphi(\mathcal{S}_{obs})$ is also a real number. In the frequentist tradition, one then studies properties of the sampling distribution of the statistic $\varphi(\mathcal{S})$ assuming random sampling. We did this previously when considering the standard error of the generalized sample mean in a previous subsection.

However, when nontrivial RVVMs are observed, the random sampling mechanism is not the only one that generates uncertainty in the distribution of $\varphi(\mathcal{S})$. Indeed, if $\mathcal{S}$ is a random sample consisting of generic RVVMs, then the distribution of the random variable $\varphi(\mathcal{S})$ is governed both by the random sampling structure and by the RVVMs themselves; we write $\varphi(\mathcal{S}_\rho)$ to emphasize this joint stochastic dependency. Now, when one fixes an observed sample of $n$ measurements, say $\mathcal{S}_{\rho,n} = \{\mu_1, \ldots, \mu_n\}$, the sample statistic $\varphi(\mathcal{S}_{\rho,n})$ is still a random variable, with distribution given by the joint measure induced by the collection of observed RVVMs, $\{\mu_1, \ldots, \mu_n\}$. The natural way to then convert this sample statistic into an actual point estimator is to take an expectation over this joint measure; i.e., we define the *generalized point statistic*:

$$\mathrm{AA}(\varphi_\rho) := \int_{\mathbb{R}^n} \varphi(\boldsymbol{x}) \; d\mu_{\otimes_n}(\boldsymbol{x}), \tag{10}$$

where $\mu_{\otimes_n}$ denotes the joint probability measure generated by the sample data, $\mathcal{S}_{\rho,n}$. If $\varphi : \mathbb{R}^n \to \mathbb{R}$, then this is a a real-valued sample statistic defined by averaging all possible values of the ordinary point statistic over all hypothetical fixed point observations defined by the observed RVVMs; i.e., the generalized statistic is an *average over arrangements* (hence, the

"AA" notation), where each fixed data arrangement is weighted by its plausibility of occurring given by the RVVMs. We formalize the idea of an *arrangement* in the following definition.

**Definition 6**. *Let $\{\mu_1, \ldots, \mu_n\}$ be an observed sample of n RVVMs generated by a common measurement protocol. An arrangement is a set $\mathbf{z} = \{z_1, \ldots, z_n\} \subset \mathbb{R}^n$ such that $z_i \in supp(\mu_i)$ for all $1 \leq i \leq n$.*

Note that $\mathbf{Z} \sim \mu_{\otimes_n}$ defines a random variable on $\mathbb{R}^n$ and any random draw from this random variable is a random arrangement. This will come in handy when establishing some mathematical details in Example 9 and in the last subsection of this paper when we define a generalized likelihood function.

It's important to realize here that $\varphi$ is still the main statistic under consideration, whereas the AA-estimator $AA(\varphi_\rho)$ is a derived statistic that always yields a point estimate. Note the conceptual similarity between this framework and the Bayesian tradition: One constructs a posterior distribution for a target quantity of interest (akin to $\varphi$) and then creates a point estimator out of this distribution by, say, taking an expectation with respect to this distribution (akin to the AA-estimator). This is just a conceptual similarity right now, but we will return to Bayesian estimation with RVVMs in the next subsection.

When $\varphi(\boldsymbol{x})$ is the sample mean, this is the previously defined estimator $AA(\overline{X}_\rho)$ considered in (8). When $\varphi(\boldsymbol{x})$ is the classical unbiased sample variance estimator, (10) becomes

$$AA(s_\rho^2) := \int_{\mathbb{R}^n} \frac{1}{n-1} (x_i - \overline{x})^2 \ d\mu_{\otimes_n}(\boldsymbol{x}), \tag{11}$$

where we understand the notation $\overline{x} = \frac{1}{n} \sum_{i=1}^{n} x_i$ for any given hypothetical arrangement $\boldsymbol{x}$. We already saw in a previous subsection that the AA-estimator of the mean equals the original one we defined in (6), whose definition was motivated by the idea of first order Berkson calibrated RVVMs to yield unbiased estimation of the mean of $X$. In the previous subsection, we saw that the generalized sample variance statistic $S_\rho^2$ of (9) was an unbiased estimator of the variance of $X$ under first and second order Berkson calibration. It is interesting to observe now the following. (The proof of this proposition appears in S1 File).

**Proposition 4**. *The AA-estimator of the variance, $AA(s_\rho^2)$, is not in general equal to $S_\rho^2$, assuming a Berkson calibrated structure or otherwise, and so is not necessarily an unbiased estimator of the variance of X. However, the AA-estimator is still asymptotically unbiased.*

AA-estimators are attractive since they give a natural way of generalizing any point statistic of interest to the general sample data setting of nontrivial RVVMs. The AA-estimator of the mean is always unbiased, and the AA-estimator of the variance, while only asymptotically unbiased, seems to not disagree too much with the unbiased generalized sample variance $S_\rho^2$ for moderately small sample sizes and mildly informative RVVMs. For instance, a random sample of 30 RVVMs, each normally distributed with means ranging from 2 to 20 and standard deviations ranging from above 0 to 5, generate $S_\rho^2$ and $AA(s_\rho^2)$ sample estimates that differ by less than 0.01 on average, a relative difference to the size of the point estimate of less than 1 in 38,000 (see S2 File simulation code).

Furthermore, AA-estimators can be naturally approximated using Monte Carlo approaches. Computational cost could be high depending on the complexity of the RVVMs and the underlying sample size, but Monte Carlo (or other) approximations of the integral over the joint measure in (10) are at least familiar territory for applied analysts. Note too that since AA-estimators are functions of the joint measure in (10), they can naturally accommodate measurement protocols that generate non-independent sample measurement processes.

As AA-estimators collapse all measurement uncertainty encoded in the RVVMs, one can unambiguously talk about their standard errors. However, we have already seen at least one instance where this approach has felt somewhat unsatisfactory, namely, when considering the example of a measurement protocol of total ambivalence for a Bernoulli target phenomenon. Then, measurement protocols can generate statistics that vary quite little (or not at all) from one random sample to another, yet it hardly seems intuitively appropriate to declare that such statistics have little (or no) standard error. In fact, the sampling variance of the sample mean under the total ambivalence measurement protocol is zero, but there is considerable *measurement uncertainty* attached to the sample mean estimate under that measurement protocol. The way to redress this apparent disconnect is to remember and utilize the distinction between the main statistic $\varphi$ under consideration, and its derived point estimator $AA(\varphi_\rho)$.

For an AA-estimator, one can unambiguously define its standard error as

$$SE^2(AA(\varphi_\rho)) = \text{Var}_\Omega(AA(\varphi_\rho)) = \text{Var}_\Omega[\mathbb{E}_\rho(\varphi \mid \mathcal{S})], \tag{12}$$

where $\mathcal{S}$ is a random sample drawn from the sample space $\Omega$ and we have used the shorthand $\mathbb{E}_\rho(\varphi \mid \mathcal{S}) = \int_{\mathbb{R}^n} \varphi(\boldsymbol{x}) \; d\mu_{\otimes_n}(\boldsymbol{x})$. As usual, this quantity captures the sampling variance present in the point estimator; that is, *it captures how much we expect the value of the statistic to vary from one random sample to another*. This quantity does not, however, capture anything explicit about the *measurement uncertainty* generated by the RVVMs, as this probability information is averaged away via the construction of the AA-estimator itself. Note that the standard error defined in (12) also makes sense for any non-AA point estimator derived from a sample of RVVMs, for example, the generalized unbiased sample variance estimator (assuming first and second order Berkson calibration) of (9).

Sampling uncertainty is only one component of interest. For a generic statistic $\varphi$ that is a function of both a random sample $\mathcal{S}$ and a set of nontrivial RVVMs on $\mathcal{S}$, a natural quantity to consider is what could be referred to as the statistic's *total standard error*:

**Definition 7**. *Let $\varphi$ be a statistic and let $\rho$ be a measurement protocol for a target random variable $X$ defined on a sample space $\Omega$. Let $\mathcal{S}$ be a random sample from $\Omega$. Define the* **total variance** *of the statistic to be*

$$Var_{\Omega \times \rho}(\varphi) = \text{Var}_\Omega[\mathbb{E}_\rho(\varphi \mid \mathcal{S})] + \mathbb{E}_\Omega[\text{Var}_\rho(\varphi \mid \mathcal{S})], \tag{13}$$

*where the expectations over $\Omega$ are with respect to the sampling distribution of the statistic and the expectations over $\rho$ are with respect to the joint measure generated by the sample RVVMs. The* **total standard error** *of the statistic is the positive square root of the total variance.*

When the measurement protocol $\rho$ generates only trivial RVVMs (calibrated or not), the total variance collapses to only the first term of (13), which is also the naive sampling variance considered in (12). We can refer to this first component of (13) as the *classical* or *sampling variance* of the statistic. The second component of (13) could be referred to as the *measuring variance* of the statistic. The sum of these two quantities, the *total variance*, is then a natural combined measure of the sampling and measurement uncertainty of a statistic. Notice that the total variance of a statistic is the sum of the sampling variance of its corresponding AA-estimator and its measuring variance, which is literally the average of its intrinsic measurement error variance. So the total variance of an AA-estimator is simply its sampling variance, since an AA-estimator is a point estimator by definition.

It is important to realize that classical notions of measurement error only capture the type of uncertainty described in the first component of (13). This is a bit counterintuitive given the classical terminology, but can be better understood as follows. Recall that classical notions of

measurement error invariably rely on some kind of model that specifies an algebraic or likelihood-based connection between a measurand of interest $X$ and an observable proxy $X^*$, as in Eqs (1), (2) and (3). These models often allow one to decompose the sampling variance $\text{Var}_\Omega[\mathbb{E}_\rho(\varphi \mid \mathcal{S})]$ into at least two pieces, where one piece usually captures variance of the statistic assuming no (incidental) measurement error and the other quantifies additional variance due to using the noisy proxy $X^*$ in place of $X$.

**Example 7**. As a concrete example, take an example of classical measurement error where we propose $X^*|X \sim N(X, \sigma^2)$, a model of the type given by (2), and let $\varphi$ denote the sample mean. Then

$$
\begin{aligned}
\text{Var}(\overline{X^*}) &= \text{Var}[\mathbb{E}(\overline{X^*} \mid X)] + \mathbb{E}[\text{Var}(\overline{X^*} \mid X)] \\
&= \text{Var}(\overline{X}) + \sigma^2.
\end{aligned}
\tag{14}
$$

All expectations here are with respect to the usual sampling distribution of the statistic, so this is really just a decomposition of the first component of (13). Indeed, the second component of the total variance is always zero since the classical framework does not encompass uncertainty due to intrinsic measurement error. One can be very explicit about this by converting the classical measurement error model we are assuming, $X^*|X \sim N(X, \sigma^2)$, into the language of measurement protocols, yielding a trivial measurement protocol $\rho$ subject to (incidental) measurement error via the assumed parametric structure. To whit:

$$
\begin{aligned}
\text{Var}_{\Omega\times\rho}(\overline{X}_\rho) &= \text{Var}_\Omega[\mathbb{E}_\rho(\overline{X}_\rho \mid \mathcal{S})] + \mathbb{E}_\Omega[\text{Var}_\rho(\overline{X}_\rho \mid \mathcal{S})] \\
&= \text{Var}_\Omega(\overline{X^*}) + 0,
\end{aligned}
$$

since the statistic $\overline{X}_\rho$ is constant for a given sample over the joint measure it generates with $\rho$. The resulting variance is now exactly what was decomposed further in (14). This is why referring to the second term of the total variance decomposition in (13) as *measuring variance* is advisable: the first component of (13) can capture uncertainty due to incidental measurement error (which produces no uncertainty on a sample measurement), and only the second component can quantify uncertainty due to intrinsic measurement error.

A few examples are in order to further illustrate things.

**Example 8**. First, we return to the example of a target Bernoulli phenomenon $X$ with measurement protocol of total ambivalence: $\rho(\omega) \sim Ber(1/2)$ for all $\omega \in \Omega$. As previously noted, for any sample $\mathcal{S} \subset \Omega$, one has $\bar{X}_\rho = 0.5$. Since this is true of any sample, we noted that the classical standard error (theoretical or estimated) is identically zero; i.e., the sampling standard error is zero. This is just one component of the total standard error of the statistic under this measurement protocol though. Using our definition, the total standard error is the sum of two pieces, the first of which is zero. The second component firsts asks for the variance of $\overline{X}_\rho$ for a given sample $\mathcal{S} = \{\mu_1, \ldots, \mu_n\}$. These measures are all distributed as $Ber(1/2)$, so this variance is $(4n)^{-1}$. This variance is in fact constant over all samples (without replacement) of size $n$ since the measurement protocol dictates total ambivalence, so taking an expectation over the sample space changes nothing. Hence, the total standard error of $\overline{X}_\rho$ for a sample of size $n$ under this measurement protocol is simply $(4n)^{-1/2}$. As the sample sizes increases, this total standard error approaches zero, reflecting the fact that the averaging operation of the sample mean statistic concentrates the joint measure of the observed RVVMs around the point mass at 0.5; i.e., this is the concentration of measures phenomenon for the $n$-fold convolution that defines the distribution of the sample statistic.

We have already noted that the total variance of a statistic reduces to its sampling variance precisely when the relevant measurement protocols produce only trivial RVVMs. A natural

question to ask then is are there situations where the total variance would reduce to only a nonzero measuring variance? The answer is yes, though this will virtually never occur in practice. When working with data from a true census of a population, sampling variance is necessarily zero. However, sample measurements can still of course be subject to intrinsic measurement error. Consequently, statistics (i.e., functions of the data) need not conform to the true values of their estimands in the population. All statistical uncertainty would be due to measuring variance and any statistic's total variance would be exactly equal to its measuring variance. In practice, direct calculation of the theoretical total variance will usually be intractable. However, one can get an empirical approximation of this quantity simply by exploiting a familiar bootstrap and Monte Carlo technique.

**Example 9**. For example, consider the following simulation scenario (reproducible R code [34] appears in Connections with related S2 File). First, we relate $X$ and $Y$ at the population level via the following simple expression:

$$Y(\omega) = 1 + 5X(\omega) + \varepsilon(\omega), \tag{15}$$

where $\varepsilon \sim N(0, SD = 0.5)$ is a random perturbation and $X \sim N(5, SD = 3)$. Then, we observe 100 observations under various measurement protocols for $X$ and $Y$ that generate different types of calibrated RVVMs for $X$ and $Y$, some trivial (i.e., deterministic) and some non-trivial. The eleven different measurement protocols we will consider are described in Table 1 and their properties are unpacked in detail below. We use the notation $\mathrm{id}_X(\omega) = \delta_{X(\omega)}$ and $\mathrm{id}_Y(\omega) = \delta_{Y(\omega)}$ to denote deterministic measurement protocols free of measurement error. Also, to simplify notation, we let $\rho_i$ denote the $i$th measurement protocol for $X$ and let $\tau_i$ denote the $i$th measurement protocol for $Y$. Under each measurement protocol $(\rho_i, \tau_i)$, we then use our 100 sample observations $\{(\rho_i(\omega_j), \tau_i(\omega_j)) : 1 \leq j \leq 100\}$ to construct the AA-estimators of the OLS solutions to the (correctly specified) regression model

$$\tau_i(\omega) = \beta_0 + \beta_1 \rho_i(\omega) + \varepsilon(\omega), \quad \varepsilon \sim N(0, \sigma^2). \tag{16}$$

Measurement protocol 1, $(\rho_1, \tau_1)$, is deterministic and free of measurement error for both $X$ and $Y$; this is the classic data analysis scenario where sample measurements are made without

**Table 1. Eleven different measurement protocols for studying the population relationship between $X$ and $Y$ defined in (15).**

| Measurement protocol for $X$ | Measurement protocol for $Y$ | Berkson calibration? | Classical calibration? | Intrinsic meas. error? |
|---|---|---|---|---|
| $\rho_1(\omega) = \mathrm{id}_X(\omega)$ | $\tau_1(\omega) = \mathrm{id}_Y(\omega)$ | Yes | Yes | No |
| $\rho_2(\omega) = \begin{cases} \delta_{x_l}, & X(\omega) \leq 5 \\ \delta_{x_r}, & X(\omega) > 5 \end{cases}$ | $\tau_2(\omega) = \mathrm{id}_Y(\omega)$ | Yes | No | No |
| $\rho_3(\omega) = \delta_{a(\omega)}, a(\omega) \sim N(X(\omega), 2)$ | $\tau_3(\omega) = \mathrm{id}_Y(\omega)$ | No | Yes | No |
| $\rho_4(\omega) = N(X(\omega), \sqrt{v(\omega)})$ | $\tau_4(\omega) = \mathrm{id}_Y(\omega)$ | 1st order | 1st order | $X$ only |
| $\rho_5(\omega) = N(X^*(\omega), \sqrt{v(\omega)})$ | $\tau_5(\omega) = \mathrm{id}_Y(\omega)$ | No | 1st order | $X$ only |
| $\rho_6(\omega) = \begin{cases} U(x_l - v(\omega), \ x_l + v(\omega)), & X(\omega) \leq 5 \\ U(x_r - v(\omega), \ x_r + v(\omega)), & X(\omega) > 5 \end{cases}$ | $\tau_6(\omega) = \mathrm{id}_Y(\omega)$ | 1st order | No | $X$ only |
| $\rho_7(\omega) = \begin{cases} U(l_{min}, \ l_{max}), & X(\omega) \leq 5 \\ U(r_{min}, \ r_{max}), & X(\omega) > 5 \end{cases}$ | $\tau_7(\omega) = \mathrm{id}_Y(\omega)$ | 1st & 2nd order | No | $X$ only |
| $\rho_8(\omega) = U(X(\omega) - 2v(\omega), X(\omega) + 2v(\omega))$ | $\tau_8(\omega) = \mathrm{id}_Y(\omega)$ | 1st order | 1st order | $X$ only |
| $\rho_9(\omega) = U(X^*(\omega) - 2v(\omega), X^*(\omega) + 2v(\omega))$ | $\tau_9(\omega) = \mathrm{id}_Y(\omega)$ | No | 1st order | $X$ only |
| $\rho_{10}(\omega) = \rho_8(\omega)$ | $\tau_{10}(\omega) = N(Y(\omega), 10\sqrt{w(\omega)})$ | 1st order | 1st order | $Y$ and $X$ |
| $\rho_{11}(\omega) = \mathrm{id}_X(\omega)$ | $\tau_{11}(\omega) = \tau_{10}(\omega)$ | 1st order | 1st order | $Y$ only |

any kind of measurement error. Measurement protocols 2 and 3 are also fully deterministic, but measurements for $X$ are polluted by Berkson calibrated measurement error under $\rho_2$ and by classically calibrated measurement error under $\rho_3$. Note that since these measurement protocols are trivial, there can be at most one kind of calibration condition if measurement error is present, and there is no need to specify the order of the calibration, by Proposition 3.

Measurement protocols 4 through 10 all contain some kind of intrinsic measurement error for $X$. In particular, $\rho_4$ generates first order Berkson and classically calibrated RVVMs assuming a normal structure for the sample measurements; $\rho_8$ does the same except under an assumed uniform structure. The measurement protocols $\rho_5$ and $\rho_9$ are first order classically calibrated only, with the former assuming a normal structure and the latter a uniform structure for the RVVMs. Measurement protocol $\rho_6$ is first order Berkson calibrated only and $\rho_7$ is both first and second order Berkson calibrated, both assuming a uniform structure for the RVVMs. Measurement protocol 10 generates intrinsic measurement error for both $X$ and $Y$. Specifically, $\rho_{10} = \rho_8$ and $\tau_{10}$ generates normal-valued measurements for $Y$; both measurement protocols are simultaneously first order Berkson and classically calibrated. Finally, measurement protocol 11 is deterministic and free of measurement error for $X$, but polluted by first order Berkson and classically calibrated errors for $Y$, $\tau_{11} = \tau_{10}$.

Many of these measurement protocols depend on various fixed or random parameters in order to ensure the various kinds of calibration structures. The random parameters $v$ and $w$ used in measurement protocols 4 through 6 and 8 through 11 are generated by random draws from $U(0, 4)$ distributions. The random parameters $X^*$ used in measurement protocols 5 and 9 are generated by random draws from $N(X, 1)$ distributions. The fixed parameters $x_l$ and $x_r$ used in measurement protocols 2, 6, and 7 (implicitly) are the expectations of the, respectively, left and right truncated normal distributions generated by splitting $X \sim N(5, SD = 3)$ at its mean. The fixed parameters $l_{min}, l_{max}, r_{min}, r_{max}$ used in measurement protocol 7 are chosen so that the means and variances of the RVVMs agree with the mean and variance of these two truncated normal distributions.

For any given measurement protocol, the AA-estimators of the OLS solutions can be approximated via Monte Carlo simulation. Here, we created $N = 2000$ random arrangements for the $n = 100$ sample measurements generated by $\rho_i$. The AA-estimators, $\mathbb{E}_{\rho_i}(\varphi \mid \mathcal{S})$, are then approximated by the empirical average of the OLS solutions to (16) using each of these arrangements. Similarly, the (intrinsic measurement) variances of the OLS solutions on the given sample, $\text{Var}_{\rho_i}(\varphi \mid \mathcal{S})$, are approximated by the empirical variance of the OLS solutions to (16) over the $N = 2000$ arrangements. Then, to approximate the sampling variance of the AA-estimators (i.e., the sampling variance) and the sampling expectation of the variances of the OLS solutions (i.e., the measuring variance), we bootstrap this procedure over $M = 200$ iterations. Reproducible R code [34] for these calculations appears in S2 File and numerical results appear in Table 2.

The main quantities of interest are the AA-estimates of the OLS solution to the regression slope, $\beta_1$ in (16), their standard errors, and the corresponding theoretical attenuation factor. There is no theoretical attenuation of the slope towards the null for measurement protocols 1 and 2 as governed by the classical theory of deterministic measurement error, and there is also no attenuation for measurement protocol 11 where intrinsic measurement error is present in the response $Y$, but measurements for the predictor $X$ are free of all measurement error. This is anticipated by the classical theory and easy to establish mathematically for the general case of RVVMs (see these and all subsequent derivations for the attenuation factors in S1 File). One can conclude then that any kind of measurement error, incidental or intrinsic, in the response variable does not bias the OLS estimator of the predictor coefficient, it only serves to increase

**Table 2. AA-estimators, attenuation factors, and various standard errors of OLS estimates of model (16) using sample data generated from eleven different measurement protocols.**

| Measurement protocol | Estimand | AA-est. of OLS solution | Sampling Std. Error | Measuring Std. Error | Total Std. Error | Theoretical attenuation factor |
|---|---|---|---|---|---|---|
| $(\rho_1, \tau_1)$ | intercept | 1.135 | 0.090 | 0 | 0.090 | 1 |
| | slope | 4.978 | 0.016 | 0 | 0.016 | |
| $(\rho_2, \tau_2)$ | intercept | 1.023 | 1.943 | 0 | 1.943 | 1 |
| | slope | 5.055 | 0.362 | 0 | 0.362 | |
| $(\rho_3, \tau_3)$ | intercept | 8.844 | 1.365 | 0 | 1.365 | $\frac{9}{9+4} = 0.692$ |
| | slope | 3.209 | 0.213 | 0 | 0.213 | |
| $(\rho_4, \tau_4)$ | intercept | 5.352 | 0.534 | 0.971 | 1.108 | $\frac{9}{9+2} = 0.818$ |
| | slope | 4.112 | 0.105 | 0.168 | 0.198 | |
| $(\rho_5, \tau_5)$ | intercept | 7.007 | 0.641 | 1.062 | 1.240 | $\frac{9}{9+2+1} = 0.750$ |
| | slope | 3.766 | 0.124 | 0.178 | 0.217 | |
| $(\rho_6, \tau_6)$ | intercept | 6.445 | 1.685 | 1.172 | 2.053 | $\frac{9}{9+16/9} = 0.835$ |
| | slope | 3.931 | 0.312 | 0.222 | 0.383 | |
| $(\rho_7, \tau_7)$ | intercept | 9.770 | 1.371 | 1.200 | 1.822 | $\frac{9}{9+(l_{max}-l_{min})^2/12} = 0.733$ |
| | slope | 3.233 | 0.236 | 0.217 | 0.320 | |
| $(\rho_8, \tau_8)$ | intercept | 11.448 | 1.096 | 1.041 | 1.512 | $\frac{9}{9+256/36} = 0.559$ |
| | slope | 2.858 | 0.202 | 0.156 | 0.255 | |
| $(\rho_9, \tau_9)$ | intercept | 12.271 | 1.087 | 1.072 | 1.527 | $\frac{9}{9+256/36+1} = 0.526$ |
| | slope | 2.689 | 0.195 | 0.162 | 0.253 | |
| $(\rho_{10}, \tau_{10})$ | intercept | 11.458 | 1.099 | 2.481 | 2.713 | $\frac{9}{9+256/36} = 0.559$ |
| | slope | 2.855 | 0.202 | 0.385 | 0.434 | |
| $(\rho_{11}, \tau_{11})$ | intercept | 1.167 | 0.099 | 2.709 | 2.710 | 1 |
| | slope | 4.970 | 0.017 | 0.459 | 0.460 | |

the uncertainty in the estimator as evidenced by the larger total standard errors for measurement protocols 2 and 11.

For nontrivial measurement protocols, the amount of attenuation in the AA(OLS) estimator of $\beta_1$ is determined in general by the average variance of the RVVMs (see S1 File for details), and *not* by the actual distribution of the RVVMs. Comparing the output of measurement protocols 4 and 5, one sees that this attenuation is greater when Berkson calibration is absent; the same is true when comparing measurement protocols 8 and 9. In these latter cases, the average variance of the RVVMs is larger than for measurement protocols 4 and 5, so the attenuation factors are closer to zero and the total standard errors are larger.

Measurement protocol 6 is first order Berkson calibrated only, while measurement protocol 7 is both first and second order Berkson calibrated, both generating uniform-valued measurements. It is interesting to note that while the attenuation factor is closer to zero for the data generated by measurement protocol 7, the total standard error of the slope estimate is less than for the data generated by measurement protocol 6. This is to be expected as the extra order of Berkson calibration ensures a lower sampling variance, since both the expectations *and* the variances of all sample measurements must agree with the conditional expectations and variances of $X$ over the resulting fibres of the measurement protocol.

The estimators from the data generated by measurement protocols 8 and 10 are theoretically identical since they were generated using the same measurement protocols for the predictor $X$. Moreover, the corresponding sampling standard errors are also theoretically identical (the small discrepancies in the tabulated values are due to Monte Carlo error). However, measurement protocol 10 also contained intrinsic measurement error in the response $Y$, and thus

the corresponding measuring standard error (and so too the total standard error) is larger for the estimator generated by the data from measurement protocol 10.

We note briefly that all the OLS estimates of the intercept parameter in model 16 are far more inaccurate compared to the slope estimates when attenuation is present due simply to the fact that for our particular example most of the density of the predictor $X$ falls away from the intercept at $X = 0$. Centering this predictor would greatly decrease these inaccuracies.

Finally, as a graphical supplement, empirical approximations to the distributions of the OLS estimators generated under measurement protocol 4 are displayed in Fig 2. Note once again that these are the distributions of the OLS estimators on the given sample; i.e., since our sample data are random variables, so too are classical sample statistics. The variances of these distributions are quantified by the measuring variances of the statistics.

**Likelihood-based estimation.** The theory and examples of the previous section suggest a natural approach towards likelihood-based estimation, namely, by constructing AA-estimators. This is one approach, but there is at least one other equally reasonable approach to likelihood estimation that one could take. While a complete development of these likelihood-based approaches to estimation and inference is far too big a task for this paper, we take the opportunity now to formalize a bit of the foundation in this final section by expanding on the previous sections and on some of the initial development in Kroc [1].

We begin with an array of $n$ measurements, each for $p$ random variables:

$$\rho_{\mathbf{X}}(\boldsymbol{\omega}) := \{\rho_{X_j}(\omega_i) : 1 \leq j \leq p, \ 1 \leq i \leq n\},$$

where $\mathbf{X} = \{X_1, \ldots, X_p\}$ and $\boldsymbol{\omega} = \{\omega_1, \ldots, \omega_n\}$. To keep matters simple, we assume that each of the $p$ measurement protocols generate *totally mutually independent measurements*; i.e., that measurements are independent for any choice of sample unit and measurement protocol: $\rho_{X_{j_1}}(\omega_{i_1}) \perp \rho_{X_{j_2}}(\omega_{i_2})$, with at least one of $i_1 \neq i_2$ or $j_1 \neq j_2$, where $\omega_{i_1}, \omega_{i_2} \in \Omega$. In the following, we will take advantage of this simplification constantly. We denote some likelihood specification proposed on the collection of target random variables $\mathbf{X}$, parameterized by a vector of unknown constants $\boldsymbol{\theta}$, by

$$f_{\mathbf{X}}(\cdot \mid \boldsymbol{\theta}). \tag{17}$$

If one were to observe traditional deterministic measurements on the vector $\mathbf{X}$ (i.e., all measurement protocols generated trivial and calibrated RVVMs), then (17) would become a

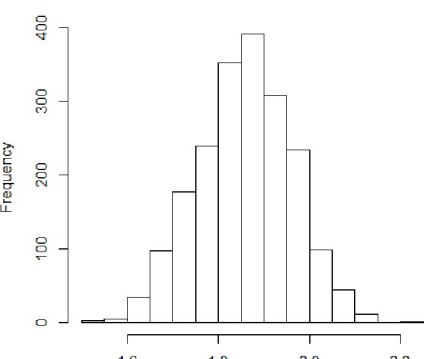 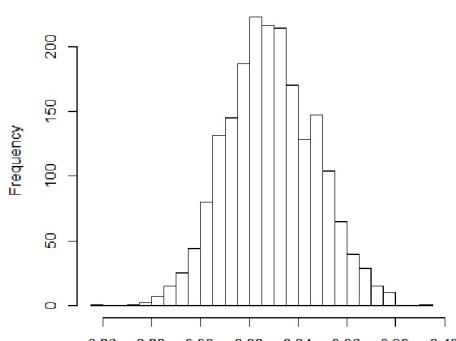

**Fig 2. Empirical approximations to the distributions of the OLS estimates of the regression coefficients in model (16) under measurement protocol 4.**

traditional likelihood function for these data on the proposed parametric model. More generally, however, we want to define a *generalized likelihood function* for the generic set of measurement protocols $\rho_\mathbf{X}$ evaluated at the sampled units $\boldsymbol{\omega} \in \Omega^n$. We formally denote this object by "plugging into" the notation of (17):

$$f_\mathbf{X}(\rho_\mathbf{X}(\boldsymbol{\omega}) \mid \boldsymbol{\theta}). \tag{18}$$

This is currently just a formal notation. Again, note that, formally at least, this reduces to a traditional likelihood function if all measurement protocols generate trivial and calibrated RVVMs. Now, our set of sample data is simply an array of Borel probability measures on $\mathbb{R}$:

$$\rho_\mathbf{X}(\boldsymbol{\omega}) = \{\boldsymbol{\mu}_j : 1 \leq j \leq p\} = \{\mu_{j,i} : 1 \leq j \leq p, \; 1 \leq i \leq n\}.$$

For each $j \in \{1, \ldots, p\}$, define the *n*-vector of random variables $\mathbf{Z}_j = \{Z_{j,1}, \ldots, Z_{j,n}\}$ where $Z_{j,i} \sim \mu_{j,i}$. Notice that for every vector $\mathbf{z}_j \in \mathbb{R}^n$, the vector $\mathbf{Z}_j(\mathbf{z}_j)$ is a fixed sequence of real numbers, and as $\mathbf{z}_j$ spans $\mathbb{R}^n$, this vector spans the range of $X_j$, one of the *p* target random variables of interest. Thus, $f_{X_j}(\mathbf{Z}_j(\mathbf{z}_j) \mid \boldsymbol{\theta})$ is an honest likelihood function. But we can bootstrap this process by exploiting the law of total probability for each target random variable $X_j$ simultaneously. Thus, we can now functionally define the *generalized likelihood function* for our selected sample units, given measurement protocols, and proposed parametric model as:

$$f_\mathbf{X}(\rho_\mathbf{X}(\boldsymbol{\omega}) \mid \boldsymbol{\theta}) = \int_{\mathbb{R}^n} \cdots \int_{\mathbb{R}^n} f_\mathbf{X}(\mathbf{Z}_1(\mathbf{z}_1), \ldots, \mathbf{Z}_p(\mathbf{z}_p) \mid \boldsymbol{\theta}) \; d\boldsymbol{\mu}_1(\mathbf{z}_1) \cdots \boldsymbol{\mu}_p(\mathbf{z}_p). \tag{19}$$

Notice that we rely on Fubini's Theorem to ensure that the order of integration in (19) is immaterial. We have already used the fact that our measurement protocols are totally mutually independent, but not to the fullest extent. We do so now by recognizing that the traditional likelihood function in the integrand of (19) can be further decomposed into a product of likelihood functions over the *n* sample units as:

$$f_\mathbf{X}(\rho_\mathbf{X}(\boldsymbol{\omega}) \mid \boldsymbol{\theta}) = \int_{\mathbb{R}^n} \cdots \int_{\mathbb{R}^n} \prod_{i=1}^{n} f_\mathbf{X}(Z_{1,i}(z_{1,i}), \ldots, Z_{p,i}(z_{p,i}) \mid \boldsymbol{\theta}) \; d\boldsymbol{\mu}_1(\mathbf{z}_1) \cdots \boldsymbol{\mu}_p(\mathbf{z}_p). \tag{20}$$

Note that when all $\mu_{j,i} = \delta_{Xj}(\omega_i)$, the generalized likelihood collapses to the traditional likelihood on a sample of *n* deterministic measurements for the *p* random variables of interest.

One can define likelihood-based estimators using the generalized likelihood in much the same way as a traditional likelihood. For instance, Kroc [1] used the generalized likelihood to consider a classical Bayes' estimator of $\theta \in \mathbb{R}$ for a single target variable $X$, defined via:

$$
\begin{aligned}
\mathbb{E}(\theta \mid \rho_X(\boldsymbol{\omega})) \;\; &= \mathbb{E}_{\boldsymbol{\mu}}[\mathbb{E}(\theta \mid \rho_X(\boldsymbol{\omega}), \mathbf{z})] \\
&= \int_{\mathbb{R}^n} \int_{\mathbb{R}} \theta \cdot f_X(\theta \mid \mathbf{Z}(\mathbf{z})) \; d\theta \; d\boldsymbol{\mu}(\mathbf{z}) \\
&= \int_{\mathbb{R}} \theta \left[ \int_{\mathbb{R}^n} f_X(\theta \mid \mathbf{Z}(\mathbf{z})) \; d\boldsymbol{\mu}(\mathbf{z}) \right] d\theta \\
&= \int_{\mathbb{R}} \theta \left[ \int_{\mathbb{R}^n} \frac{1}{N(\mathbf{Z}(\mathbf{z}))} f_X(\mathbf{Z}(\mathbf{z}) \mid \theta) \; d\boldsymbol{\mu}(\mathbf{z}) \right] \pi_0(\theta) \; d\theta,
\end{aligned}
\tag{21}
$$

for some given prior $\pi_0$ on $\theta$ and where $N(\cdot)$ denotes the usual normalizing constant. For the special case of measurement protocols for a Bernoulli phenomenon $X$, Kroc [1] showed that this estimator is an asymptotically unbiased estimator of the characteristic parameter $\Pr(X = 1)$ for a generic Beta prior when $\rho$ is first order Berkson calibrated to $X$ and, under an

additional regularizing condition on the amount of intrinsic measurement error present in the RVVMs, that it is also a consistent estimator.

The application of Fubini's Theorem, as in the derivation of (21), makes a Bayesian approach to estimation particularly attractive (and analytically tractable) for general RVVMs; however, it is of course natural to consider what a purely maximum likelihood approach to estimation might look like. We take this opportunity only to point out that there are at least two equally coherent ways one could go about constructing such a theory.

First, consider what could most naturally be called the *maximum generalized likelihood estimator* of a real-valued parameter $\theta$:

$$MGLE(\theta \mid \rho_{\mathbf{X}}(\boldsymbol{\omega})) := \max_{\varphi \in \mathbb{R}} f_{\mathbf{X}}(\rho_{\mathbf{X}}(\boldsymbol{\omega}) \mid \varphi) \tag{22}$$

When $\rho_{\mathbf{X}}$ generates only calibrated and trivial RVVMs, this reduces to the traditional MLE of $\theta$ under the proposed likelihood structure. However, there is no reason why this estimator need be the same as the second natural one we could consider, one that could most naturally be called the *average maximum likelihood estimator*:

$$AMLE(\theta \mid \rho_{\mathbf{X}}(\boldsymbol{\omega})) := \int_{\mathbb{R}^n} \cdots \int_{\mathbb{R}^n} MLE(\theta \mid Z_{1,i}(z_{1,i}), \ldots, Z_{p,i}(z_{p,i})) \ d\boldsymbol{\mu}_1(\mathbf{z}_1) \cdots \boldsymbol{\mu}_p(\mathbf{z}_p). \tag{23}$$

Again, notice that when $\rho_{\mathbf{X}}$ generates only calibrated and trivial RVVMs, this reduces back to the traditional MLE of $\theta$ under the proposed likelihood structure. Note too that the AMLE is not actually an explicit function of the generalized likelihood in (20), and that it is in fact an AA-estimator as previously defined.

Clearly, the MGLE and AMLE are two very different estimators, yet both are natural generalizations of the classical MLE to the context of general measurement protocols. We do not attempt a formal study of these estimators, their properties, and how they can be related here. We simply provide an example to show that these two generalizations can yield different sample estimators under the same proposed likelihood structure and the same observed measurements.

**Example 10**. Consider the following sample data and modelling scenario. Suppose we draw a sample of four elements from a target population: $\boldsymbol{\omega} = \{\omega_1, \omega_2, \omega_3, \omega_4\}$. On each of these sample units, we apply two different measurement protocols, one for a continuous random variable $Y$ and one for a binary random variable $X$, taking values in $\{\pm 1\}$. To keep things simple, the measurement protocol for $Y$ will generate trivial RVVMs and be free of measurement error; i.e., $\rho_Y(\omega_i) = \delta_{Y(\omega_i)}$ for all $i$. The measurement protocol for $X$ will generate affine transformations of Bernoulli-valued measurements, two trivial and free of measurement error, and two not trivial so not free of measurement error:

$$\rho_X(\omega_1) = 2Ber(\theta_1) - 1, \ \ \rho_X(\omega_2) = 2Ber(\theta_2) - 1, \ \ \rho_X(\omega_3) = \delta_{X(\omega_3)}, \ \ \rho_X(\omega_4) = \delta_{X(\omega_4)}.$$

We will use these sample data to estimate the MGLE and AMLE of the corresponding slope of the simple regression line through the origin, assuming the usual linear structure, i.e., we calculate the maximum generalized likelihood estimator and the average maximum likelihood estimator of $\beta$ assuming $Y|X \sim N(\beta X, \sigma^2)$.

We begin with the AMLE, defined in Eq (23). For our example, $n = 4$, $p = 2$, $Z_{1,i} \sim \rho_Y(\omega_i)$, and $Z_{2,i} \sim \rho_X(\omega_i)$. Also, $\boldsymbol{\mu}_1$ is just a product of point-mass measures, and $\boldsymbol{\mu}_2$ is a product of two point-mass measures and the two affinely transformed Bernoulli measures. To calculate the AMLE, we first need to find the traditional MLE of $\beta$ assuming any fixed values for $Z_{1,i}, Z_{2,i}$, and then average these quantities according to the given measures. To that end, fix any $z_{1,i}, z_{2,i} \in \mathbb{R}$ and notice that since our model proposes $Y|X \sim N(\beta X, \sigma^2)$, the classical likelihood

for $\beta$ associated with $Z_{1,i}(z_{1,i})$, $Z_{2,i}(z_{2,i})$ is:

$$f(\beta \mid Z_{1,i}(z_{1,i}), Z_{2,i}(z_{2,i})) = \left(\frac{1}{\sqrt{2\pi}\sigma}\right)^4 \exp\left(-\frac{1}{2\sigma^2}\sum_{i=1}^4 (Z_{1,i}(z_{1,i}) - \beta Z_{2,i}(z_{2,i}))^2\right). \quad (24)$$

Finding the MLE of $\beta$ here is of course equivalent to maximizing the log-likelihood which, after some simplification, yields the usual MLE:

$$MLE(\beta \mid Z_{1,i}(z_{1,i}), Z_{2,i}(z_{2,i})) = \frac{\sum_{i=1}^4 Z_{1,i}(z_{1,i})Z_{2,i}(z_{2,i})}{\sum_{i=1}^4 Z_{2,i}(z_{2,i})^2}.$$

Now, to construct the AMLE, just note that for our sample data the iterated integral in (23) reduces down to a sum of four terms corresponding to the different possible values of $Z_{2,1} \times Z_{2,2} \in \{\pm 1\}^2$. That is,

$$AMLE(\beta \mid \rho_Y(\boldsymbol{\omega}), \rho_X(\boldsymbol{\omega})) =$$
$$\sum_{\substack{z_{2,1}\in\{\pm 1\}\\z_{2,2}\in\{\pm 1\}}} \frac{y_1 z_{2,1} + y_2 z_{2,2} + y_3 x_3 + y_4 x_4}{z_{2,1}^2 + z_{2,2}^2 + x_3^2 + x_4^2} \Pr(Z_{2,1} = z_{2,1})\Pr(Z_{2,2} = z_{2,2}), \quad (25)$$

where we use the classical shorthand $y_i = Y(\omega_i)$, $x_i = X(\omega_i)$ for deterministic measurements.

Now consider the maximum generalized likelihood estimator defined by Eq (22). This is a function of the generalized likelihood (20), which itself is a weighted average of traditional (hypothetical) likelihoods. Using (24) and simplifying, we find

$$MGLE(\beta \mid \rho_Y(\boldsymbol{\omega}), \rho_X(\boldsymbol{\omega})) = \max_{\phi\in\mathbb{R}}\left[\left(\frac{1}{\sqrt{2\pi}\sigma}\right)^4 \exp\left(-\frac{1}{2\sigma^2}\left((y_3 - \phi x_3)^2 + (y_4 - \phi x_4)^2\right)\right)\times\right.$$
$$\left.\sum_{\substack{z_{2,1}\in\{\pm 1\}\\z_{2,2}\in\{\pm 1\}}} \exp\left(-\frac{1}{2\sigma^2}\left((y_1 - \phi z_{2,1})^2 + (y_2 - \phi z_{2,2})^2\right)\right)\Pr(Z_{2,1} = z_{2,1})\Pr(Z_{2,2} = z_{2,2})\right] \quad (26)$$

Again, note that the AMLE is a weighted average of traditional MLEs, whereas the MGLE is the maximum of a weighted average of likelihood functions. Note too that the MGLE of $\beta$ depends on the model's other unknown parameter, $\sigma$, and that this maximum cannot be easily computed simply by switching to a log-likelihood structure; i.e., the log-generalized-likelihood offers no simplification. This is in stark contrast to the AMLE, where the freedom between the two parameters is inherited from the classical MLE of $\beta$.

It is worthwhile to examine the practical differences between these two estimators by plugging in some actual sample observations for our toy data scenario. To this end, we set

$$\rho_Y(\omega_1) = \delta_{-0.5}, \ \rho_Y(\omega_2) = \delta_{0.5}, \ \rho_Y(\omega_3) = \delta_{-1}, \ \rho_Y(\omega_4) = \delta_1,$$

and

$$\rho_X(\omega_1) = 2Ber(\theta_1) - 1, \ \rho_X(\omega_2) = 2Ber(\theta_2) - 1, \ \rho_X(\omega_3) = \delta_{-1}, \ \rho_X(\omega_4) = \delta_1,$$

and consider different values for $\theta_1, \theta_2$, corresponding to different sample uncertainties in whether $X(\omega_1)$ and $X(\omega_2)$ are equal to either $\pm 1$. The AMLEs and MGLEs for a range of different values are recorded in Table 3.

First, note that when $\theta_1, \theta_2 \in \{0, 1\}$, all RVVMs are trivial (the top and bottom rows of Table 3); consequently, the AMLE and MGLE reduce down to the ordinary maximum likelihood estimator of the slope parameter $\beta$ corresponding to the fitted model using a set of

**Table 3. The AMLEs and MGLEs for the slope parameter in the model $Y|X \sim N(\beta X, \sigma^2)$ using the toy dataset on four sample measurements for the given measurement protocols for $X$ and $Y$, and different RVVMs for $\rho_X(\omega_1)$ and $\rho_X(\omega_2)$.**

| $\theta_1$ | $\theta_2$ | AMLE | MGLE, $\sigma = 0.3$ | MGLE, $\sigma = 0.4$ | MGLE, $\sigma = 0.5$ | MGLE, $\sigma = 1$ | MGLE, $\sigma = 5$ |
|---|---|---|---|---|---|---|---|
| 1 | 0 | 0.25 | 0.25 | 0.25 | 0.25 | 0.25 | 0.25 |
| 0.9 | 0.1 | 0.3 | 0.749 | 0.699 | 0.452 | 0.316 | 0.301 |
| 0.8 | 0.2 | 0.35 | 0.75 | 0.730 | 0.627 | 0.384 | 0.351 |
| 0.7 | 0.3 | 0.4 | 0.75 | 0.739 | 0.684 | 0.451 | 0.402 |
| 0.6 | 0.4 | 0.45 | 0.75 | 0.743 | 0.710 | 0.513 | 0.452 |
| 0.5 | 0.5 | 0.5 | 0.75 | 0.745 | 0.724 | 0.569 | 0.503 |
| 0.4 | 0.6 | 0.55 | 0.75 | 0.747 | 0.733 | 0.618 | 0.553 |
| 0.3 | 0.7 | 0.6 | 0.75 | 0.748 | 0.739 | 0.659 | 0.603 |
| 0.2 | 0.8 | 0.65 | 0.75 | 0.749 | 0.744 | 0.695 | 0.652 |
| 0.1 | 0.9 | 0.7 | 0.75 | 0.749 | 0.747 | 0.724 | 0.701 |
| 0 | 1 | 0.75 | 0.75 | 0.75 | 0.75 | 0.75 | 0.75 |

The AMLEs are computed by plugging into Eq (25); the MGLEs are computed by numerically solving Eq (26) to three decimal places. Graphs of some selected generalized likelihoods are viewable in S3 File.

fully deterministic sample measurements. Next, notice that the values of the AMLE are quite intuitive. That is, for intermediate degrees of uncertainty about the values of $X(\omega_1)$, $X(\omega_2) \in \{\pm 1\}$, the AMLE of $\beta$ yields a point estimate linearly intermediate in value between the ordinary MLE that would result from observing the deterministic sample measurements $X(\omega_1) = 1$, $X(\omega_2) = 0$ (top row), or $X(\omega_1) = 0$, $X(\omega_2) = 1$ (bottom row). This is a natural reflection of the intrinsic measurement error in these two sample Bernoulli-valued measurements.

The behaviour of the MGLE is more complex (visual plots of some generalized likelihoods for selected values of $\rho_X(\omega_1)$, $\rho_X(\omega_2)$, and $\sigma$ can be found in S3 File). Values are still intermediate between the deterministic endpoints, but the contribution of sample unit uncertainty to the point value of the estimator is more opaque. Moreover, as $\sigma \to \infty$, we see that the MGLE of $\beta$ seems to converge to the AMLE. This makes intuitive sense since allowing $\sigma$ to be large means that we are not interested in imposing any conditional variance structure at the conditional sample measurement level. The probability structure of the nontrivial RVVMs is all that drives the MGLE of $\beta$, and this is precisely all that drives the AMLE of $\beta$ for the proposed conditional normal model relating $X$ to $Y$.

On the other hand, one will notice that as $\sigma \to 0^+$, the MGLE approaches the value of the MLE when $X(\omega_1) = 0$, $X(\omega_2) = 1$ whenever the RVVMs are not all trivial. At first glance, this seems like a sure sign that something is wrong, but the resolution lies in the fact that the MGLE of $\beta$ depends on the value of the model's residual variance, $\sigma^2$. Requiring that $\sigma \to 0^+$ means that we are then finding an estimate for the slope $\beta$ while simultaneously requiring the variance of $Y|X$ to be as small as possible. Given the structure of the deterministic sample observations, $(x_3, y_3) = (-1, -1)$, $(x_4, y_4) = (1, 1)$, minimal conditional variance is achieved when $Z_{2,1} = -1$, $Z_{2,2} = 1$, since we also have $y_1 = -0.5$, $y_2 = 0.5$. The situation is graphically explained in Fig 3. Here, we see that the first arrangement of points in panel (a) yields minimum conditional sample variance over all values of $Z_{2,1}$ and $Z_{2,2}$ with joint nonzero probability. It is this hypothetical set of deterministic observations (hypothetical only because two of our sample measurements are not deterministic) that would best "fit the model" $Y|X \sim N(\beta X, \sigma^2)$, with $\sigma$ as small as it can be over the arrangements with nonzero probability of the four panels of Fig 3.

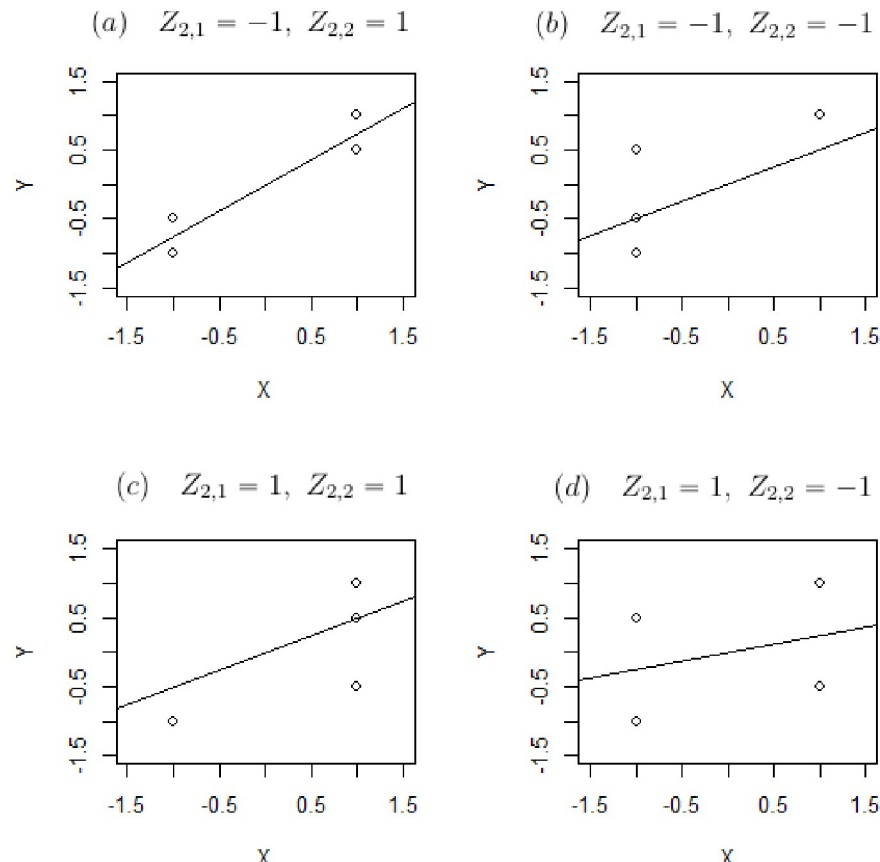

**Fig 3. Fixed data arrangements with nonzero probability for our sample RVVMs.** Since the nontrivial RVVMs denoted by $Z_{2,1}$ and $Z_{2,2}$ are affine transformations of Bernoulli random variables, there are only four joint values of $\{Z_{2,1}, Z_{2,2}\}$ with nonzero probability. Each panel above displays one of these four possible arrangements plotted as $Y$ versus $X$. The best fitting line through the origin according to MLE has been included in each instance. Note that the residual variance will be minimal for the arrangement in panel (a).

One notices an ambiguity now in the definition of the MGLE in Eq (22): Should the estimate be conditional on fixed values of other potential unknown parameters, or should the maximization occur simultaneously over all unknowns? Table 3 describes the conditional approach for our example, but one could just as easily consider what the MGLE of $\beta$ would be when maximizing the generalized likelihood simultaneously in $\beta$ and $\sigma$. The results are summarized in Table 4.

Note again that the MGLEs of both unknowns coincide with the traditional MLEs in the top and bottom rows (i.e., with fully deterministic sample data). More generally, we see that the joint MGLEs seem to often be identifying the arrangement of sample observations depicted in Fig 3, panel (a). The pull of the joint MGLEs toward this "optimal" arrangement for the proposed model is strong, as the corresponding MLEs are identified even when the panel (a) arrangement of hypothetical deterministic observations is less likely than other arrangements, as when $\theta_1 = 0.6$ and $\theta_2 = 0.4$. This is not universally so though, as evidenced by the first few rows of Table 4. There, the low probability of the "optimal" arrangement of $Z_{2,1} = -1, Z_{2,2} = 1$ is enough to drag the joint MGLEs toward the opposing arrangement of $Z_{2,1} = 1, Z_{2,2} = -1$, given by the table's first row. In this sense, the joint MGLE balances uncertainty in the sample measurements (i.e., the distributions of the RVVMs) and the maximum likelihood fit of the

**Table 4. The joint MGLEs of $\beta$ and $\sigma$.**

| $\theta_1$ | $\theta_2$ | $MGLE(\beta)$ | $MGLE(\sigma)$ |
|---|---|---|---|
| 1 | 0 | 0.25 | 0.75 |
| 0.9 | 0.1 | 0.339 | 0.714 |
| 0.8 | 0.2 | 0.472 | 0.623 |
| 0.7 | 0.3 | 0.75 | 0.252 |
| 0.6 | 0.4 | 0.75 | 0.25 |
| 0.5 | 0.5 | 0.75 | 0.25 |
| 0.4 | 0.6 | 0.75 | 0.25 |
| 0.3 | 0.7 | 0.75 | 0.25 |
| 0.2 | 0.8 | 0.75 | 0.25 |
| 0.1 | 0.9 | 0.75 | 0.25 |
| 0 | 1 | 0.75 | 0.25 |

proposed model, characterized by its residual variance, over all possible hypothetical deterministic measurements that are allowable given the observed RVVMs.

Finally, note that one can use the above example to compare what would happen if we examined a measurement protocol for $X$ subject to incidental measurement error. For instance, we can set $X(\omega_1) = -1$ and $X(\omega_2) = 1$, and then define a trivial measurement protocol that yields

$$\rho'_X(\omega_1) = \delta_1, \ \ \rho'_X(\omega_2) = \delta_{-1}, \ \ \rho'_X(\omega_i) = \delta_{X(\omega_i)} \text{ for } i = 3, 4.$$

This measurement protocol defines an error-prone proxy $X^*$ for $X$ subject only to incidental measurement error that mismeasures the value for $X$ only at $\omega_1$ and $\omega_2$. Since the measurand is binary, $X^*$ cannot be either Berkson or classically calibrated to $X$, unlike when the measurement protocol generates intrinsic measurement error [1]. The maximum likelihood estimator of $\beta^*$ for the proposed regression model $Y|X^* \sim N(\beta^* X^*, \sigma^2)$ is 0.25, given by the first row of Table 3, while the desired unbiased maximum likelihood estimator of $\beta$ for $Y|X \sim N(\beta X, \sigma^2)$ is 0.75, given by the last row of Table 3. One sees then that allowing the measurement protocol to encode intrinsic measurement error can only improve the accuracy of this point estimate, using either the AMLE or the MGLE. That is, allowing the deterministic mismeasurements at $\omega_1$ and $\omega_2$ to be replaced by nondegenerate Bernoulli-valued measurements ensures that the observed sample data are closer (in the distributional sense) to the unobserved true values of the measurand. Because of this, for binary measurands in particular (and categorical measurands more generally), the potential gains of moving to the general RVVM framework are potentially quite attractive.

This is only the beginning of the development of a properly generalized likelihood theory for nondeterministic data arising from general measurement protocols. However, it should hopefully now be evident that the work to be done is not simply bookkeeping; genuinely new mathematical objects and structures demand to be compared, contrasted, and understood, even for the simple case of Bernoulli-valued measurements.

## Discussion

The general theory of RVVMs proposed in this paper allows one to integrate measurement uncertainty that is unique to each sample observation into any standard data analysis via the use of generalized statistics (i.e., random variables derived from sample data subject to both sampling uncertainty and measuring uncertainty), AA-estimators (to obtain point estimates

from generalized statistics), Berkson and classical calibration conditions (to obtain unbiased point estimators), total variances (to quantify both sampling and measuring uncertainty for a generalized statistic), and generalized likelihoods (to augment the likelihood theory based on an AA-estimator approach to maximum likelihood). This theory applies to the study of any real-valued random variables and collapses to the traditional methods (i.e., standard errors based on sampling variances, Berkson and classical calibration for unbiased estimation in the presence of incidental measurement error, and ordinary maximum likelihood or Bayes' estimation of likelihood-based models) in the absence of intrinsic measurement error.

## Limitations

There are four major difficulties with using this general apparatus: (1) One needs to collect sample data that somehow encode intrinsic measurement error, (2) quantification of measurement uncertainty for generalized statistics due to intrinsic measurement error is currently computationally expensive, (3) development of software packages for easy implementation of RVVM-based data analysis is needed, and (4) making inferences and describing substantive statistical conclusions are more detailed endeavours. The second and third items here are true limitations (though hopefully temporary), while the first and fourth mostly require a rethinking of how statistical analyses and study design are commonly construed and conducted.

It is not difficult to imagine situations where one can naturally record intrinsic measurement error in a data collection process (see the Introduction, further examples below, and the examples of Kroc [1]). However, this requires a researcher to build in a mechanism to record the extra source of error *before* they go out and collect the data. The whole point of developing this theory of RVVMs is to allow one to use more information about the measurand on the sampled units than simply a forced, single-valued response. More information about what we are measuring is certainly a good thing for empirical work, but we can only use it if we record it. In some instances, this is quite straightforward; for instance, asking someone to provide a percentage level of confidence to supplement a traditional binary 0/1 response automatically defines a Bernoulli-valued measurement that then encodes this extra confidence information to account for intrinsic measurement uncertainty. In other cases, however, the method of eliciting this extra information from a sample unit is not so straightforward. Consider a survey question asking respondents to record their level of approval of the job being done by their current head of state, from total disapproval to total approval. Traditionally, a response would be encoded in either a Likert response format (often 5, 7, or 11 response categories) or an analogue scale (e.g., a slider spanning values from 0% to 100%). Such a question is sure to contain substantial intrinsic measurement uncertainty for most respondents, as most people will likely approve of only *some* aspects of the job being done by their current head of state. Classical response formats that force a deterministic response can only capture some kind of "typical" approval rating for each individual. By moving to a response format that allows a respondent to encode their uncertainty in a response, the different dimensions and interpretations of approve/disapprove can be at least partially encoded and used for subsequent analysis (and, importantly, the extra source of uncertainty due to the imprecise object of the question can be captured). However, should one ask respondents to provide a *range* of scores on the given scale? Should one further require respondents to *weight* the set of responses in their recorded range according to their confidence in each response? Should this be done on an analogue or a Likert scale? Should one instead ask respondents to literally *draw* the probability density/mass function that defines their RVVM? Clearly, these different options (and others) place different cognitive loads on the respondent, and it is not at all clear which option is ideal, either for this particular survey item or in general.

The issue of computing total variance is currently a true limitation. The method proposed in this paper to quantify the total variance of a generalized statistic relies on a Monte Carlo approximation nested within a bootstrap. It is well known that bootstrapping can be computationally burdensome for even moderately sized datasets and the fact that each step of the bootstrap here relies on an MC approximation greatly increases this burden. Future work will have to focus on deriving analytical expressions (asymptotic or otherwise) for common families of statistics over different data structures subject to different kinds of intrinsic measurement error (i.e., different families of measures generated by the measurement protocol). In [1], analytical expressions for the total variance of a proportion-based statistic were derived for Bernoulli-valued measurements using a classical Bayes' estimator; much more analytical work in this direction is necessary to facilitate practical integration of the extra information provided by nontrivial RVVMs into more general research problems.

The lack of easy-to-use software for data analysis with RVVMs is the biggest current impediment to straightforward implementation of the RVVM framework. The R code presented in this paper and in [1] provide a simple basis for implementation and can be adapted to a variety of novel research scenarios, but there is a substantial need for the development of stand-alone R packages that can interface with data consisting of nontrivial RVVMs and with other common R packages for universal application. Addressing the previous issue of overreliance on bootstrapping techniques would greatly help streamline software development initiatives.

Incorporating the extra source of uncertainty due to intrinsic measurement error further requires an updated way of thinking about uncertainty in statistical problems. While the role of uncertainty in a statistic due to sampling error is well understood and communicated in current research, the concept of additional uncertainty due to intrinsic measurement error is new. Actually applying the general RVVM framework in a diverse variety of applied research scenarios is likely the best way to start to familiarize theoreticians and applied researchers with the existence and implications of this extra source of uncertainty.

## Examples of useful applications of RVVMs

The author of this paper is currently largely focused on the task of applying the general RVVM framework to solve a variety of real world research problems. Here, I give a brief overview of the most developed applications (see [1] for other hypothetical examples).

**Kinesiology.** This application was already discussed in the Introduction. Here, the RVVM framework is employed to quantify rater uncertainty when assigning binary scores to each of the 50 different components of the Test of Gross Motor Development, a common clinical tool designed to assess motor skill development and competence in children. The classical binary scoring system is replaced here with the collection of Bernoulli-valued measurements.

**Forest restoration and meta-analysis.** While situated within the research domain of forest conservation and management, this application illustrates a broader use for RVVMs within the context of meta-analysis. When extracting information on potential moderating variables from constituent studies for use in a meta-analysis, it is not uncommon for some studies to report incomplete or imprecise information. Here, for instance, a moderator of interest on the overall success of forest regeneration and health after logging has ceased is the amount of time the tract of land has been left undisturbed to regenerate (either passively or through active restoration efforts). However, some studies will report this length of time precisely (e.g., 17 years), whereas others will only give a range (e.g., 10 to 20 years). Such a problem frequently arises within the context of meta-analysis and the usual course of action is to just use a minimum, maximum, or average of the uncertain age range (e.g., [35, 36]). Of course, such a procedure ignores the intrinsic measurement error inherent to the application, and thus

mischaracterizes any resulting standard errors. Usually, there is no apriori reason to prefer any particular value within the reported time range; thus, a continuous or discrete uniform-valued measurement can be used to encode the sample measurement specific uncertainty for the subsequent meta-regression.

**Psychology.** Here, the application is in the context of a study examining the relationship between regular exercise and psychological affect in healthy individuals. Respondents completed a series of questionnaires during the course of completing different pre-assigned exercise regimens, with questions designed to measure various features of their psychological affect (i.e., experience of feeling and mood). Respondents recorded their answers using a 7-point Likert format for these items and then also recorded how confident they were in these self-assigned scores using another 7-point Likert response format. Many different measurement protocols could be defined to reasonably combine these two sources of information, and this study will explore several of the most obvious ones. The confidence measure could be used to define a variance or entropy of the underlying RVVM, and this RVVM could be conveniently encoded as either a categorical-valued measurement (discrete on the 7 Likert categories) or a beta-valued measurement (continuous over the latent continuum of agreement with the survey item).

**Education.** This application was introduced generally in the Introduction. The idea here is to allow students to record their level of confidence in answers to true/false and multiple-choice quiz questions in an introductory statistics course. Some questions will be designed to have one correct answer while others may have multiple correct answers. Rather than students simply identifying the option(s) that they think are correct, they will be able to include a measure of confidence in their answers, with a separate level of confidence (measured on a 0% to 100% scale) for each response category in each question. Certain questions will sometimes be repeated on subsequent quizzes to assess retention of knowledge and to track if confidence in attained knowledge increases with longer exposure to the salient concepts. In this way, one can distinguish between when a student can correctly guess or infer the correct answer based on partial knowledge and when a student confidently knows the correct answer based on a mastery of the concept. Further, RVVMs will be employed on the instructor-side of the education equation: When marking written short answer responses to quiz questions, instructors will be able to assign RVVM-based scores if they are uncertain which mark is most appropriate based on a predetermined rubric that assigns a point total for student responses with different degrees of content quality. Each of these response and scoring systems will generate categorical-valued measurements. Notably, the RVVMs generated by the instructors should theoretically obey a Berkson calibrated structure since they are generated by trained experts.

**Avian ecology.** In the study of rooftop-nesting seabirds, it is sometimes difficult to definitively decide if a particular rooftop houses a nest due to restricted access to study structures and/or limited sight-ranges from nearby obstructions [37]. To quantify observer uncertainty when identifying putative nesting locations, a Bernoulli-valued measurement can be used. Presumably, these measurements will be Berkson calibrated to the measurand given these observers are experienced professionals, and in fact this can be empirically validated by checking the structure of the Bernoulli-valued measurements against follow-up measurements where definitive judgments were later able to be made (e.g., later access to the rooftop in question). Furthermore, it is often of interest to identify the exact location of the recorded nest or nests within the structure of the rooftop for more detailed spatial analysis (e.g., internest distances). When exact spatial locations cannot be determined, a Borel measure on $\mathbb{R}^2$ can instead be defined to encode the nonzero region of possible nest location (perhaps with nonuniform mass over this region). An extension of the measurement protocol and RVVM framework

proposed here to $\mathbb{R}^n$-valued measurands can then be employed to coherently synthesize this uncertain measurement information for subsequent analysis.

## Connections with related statistical ideas

The idea of RVVMs bears many conceptual similarities to a variety of already established statistical ideas. We have already shown how RVVMs directly generalize the traditional notion of measurement error in statistics (what we have called incidental measurement error), how various important calibration conditions can be generalized to the RVVMs framework, and how these conditions can be used to construct unbiased estimators of some important target quantities of common inferential interest. In these ways, RVVMs represent a natural extension of the classical study of measurement error to a domain where sample data are not assumed to necessarily be deterministic and real-valued.

As noted in the Introduction, the next most closely related technique comes from cognitive psychology, where the idea of *confidence weighting* has a long history (see [6, 7]). Confidence weighting typically requires respondents to assign a number between 0 and 100 to denote their confidence in their response to a given binary item. Then, rather than comparing sum-scores based on the raw 0/1 responses to a set of items, a weighted sum-score is computed by simply multiplying each 0/1 response by its corresponding weight (taken as a percentage). The reader will recognize this as precisely the generalized sample mean (or AA-estimator) for a set of Bernoulli-valued measurements. The confidence weighting literature however does not integrate the confidence information and the 0/1 "best guess" response into a single sample measurement; i.e., a Bernoulli random variable. Researchers in this domain have been able to examine these weighted sum-scores and study how the confidence information corresponds to correct responses; they have also defined a notion of calibration that is identical to the notion of generalized Berkson calibration introduced in this paper applied to the special case of Bernoulli-valued measurements. However, because the binary response and confidence information are considered as separate observations (i.e., as sample information on two different measurands), researchers in this domain have not been able to integrate their confidence weights into more sophisticated data analyses (e.g., regressions) or to properly quantify the total uncertainty (i.e., sampling variance and measuring variance) attached to their weighted sum-scores. The RVVM framework allows one to apply the idea of confidence weighting in a mathematically and statistically coherent way to all manners of analyses, and it also allows one to generalize the notion of confidence weighting beyond simple binary items.

The framework of RVVMs has some commonalities with the field of fuzzy numbers and fuzzy statistics [38, 39]. Commonly in applied practice, fuzzy numbers are used to combine a best estimate (like a sample mean) and a range of plausible values (like a confidence interval) into a single mathematical quantity for further analysis (see e.g. [40, 41]). The theory of triangular numbers in particular has been frequently employed to define an arithmetic of confidence intervals that generalizes the usual arithmetic applicable only to point estimators. Formally, fuzzy numbers extend the notions of real number to real-valued function and of real-valued random variable to set-valued function. This allows for the encoding of extra sources of information, not unlike how we allow sample measurements to have distributions defined by any real, Borel measure. It would be interesting to more deeply study the differences and similarities between RVVMs and fuzzy numbers, but only the broad conceptual similarities have been currently noted.

The idea of encoding qualitative information from an expert assessor relevant to some measurand has previously been explored within the context of statistical elicitation (see e.g. [26]). In this framework, the goal has often been to elicit subjective information from domain experts

to better calibrate hyperparameters of informative priors for more accurate and precise Bayesian inference. Questions as to how to best elicit such information from real experts and then convert it into useable numbers, like means and variances for informative priors, have a long history in the statistical elicitation literature [27, 28]. This framework does not consider how to encode subjective information at the sample measurement level, nor is it concerned with measurement error in general.

Missing data problems also share some conceptual similarities with RVVMs, as both are concerned with situations where a definitive measurement has not occurred. However, missing data techniques are only ever concerned with the situation where a sample measurement has failed to occur, i.e., nonresponse [42], whereas RVVMs consider the situation where a sample measurement is observed but one cannot definitively assign a single real number to that sample measurement. Moreover, missing data techniques invariably rely on the specification of some kind of statistical model to fill in the gaps in the data, while RVVMs demand a different kind of encoding of the sample measurement itself (i.e., a real Borel measure, not just a real number) that is, ideally, model-free.

## Conclusion

In this paper, we expanded on the initial work of Kroc [1] to generalize the notion of measurement error for deterministic sample data to accommodate sample measurements that contain some kind of inherent uncertainty or undecidability, what we call intrinsic measurement error. These are (nontrivial) random-variable-valued measurements (RVVMs), and they arise naturally out of a formalization of a generic measurement process via the mathematical concept of a measurement protocol. RVVMs have the ability to encode both deterministic sources of measurement error (incidental) and those more nebulous sources that could arise from the suboptimality of a measurement procedure, or from the very nature of the target of measurement itself. The latter situation occurs frequently when the measurand is not a well or objectively defined quantity, like biodiversity or anxiety. We defined calibrating conditions that generalize both classical and Berkson measurement error models to the RVVM framework, and discussed how Berkson calibration mathematicizes what it means to be an expert assessor or rater for a measurement process. Finally, we explored how point estimation and inference could proceed in this more general framework, notably introducing the class of AA-estimators, the notions of measuring variance and total standard error, and the idea of the generalized likelihood function for a given model and set of generic RVVMs.

Clearly, much work remains to be done to better understand how estimators can behave given different kinds of observed RVVMs. This work should be largely mathematical in nature, but we should also look to real applications to guide us in most effectively tackling the work. The main concept behind RVVMs, that *sample measurements* can be inherently uncertain quantities, is one that is unfamiliar to many technical people working in statistics. Hopefully, as more applications emerge that use the RVVM framework, this unfamiliarity will start to dissipate and our science will improve as we begin to incorporate sample-unit-specific sources of uncertainty into our estimates and inferences.

## Supporting information

**S1 File. Proof of Proposition 4 and derivations of theoretical attenuation factors in Table 2 for Ex. 9.**
(PDF)

**S2 File. R code for in-text simulations.**
(PDF)

**S3 File. Some selected generalized likelihoods for the toy dataset and proposed model of Example 10 follow.**
(PDF)

## Acknowledgments

The author would like to thank Malabika Pramanik for insightful comments on this work. These comments directly contributed to the clarity of formalism for general measurement protocols in particular. The author would also like to thank Bruno D. Zumbo and Oscar L. Olvera Astivia for helpful conversations on many of the ideas that appear in this paper.

## Author Contributions

**Conceptualization:** Edward Kroc.

**Data curation:** Edward Kroc.

**Formal analysis:** Edward Kroc.

**Investigation:** Edward Kroc.

**Methodology:** Edward Kroc.

**Project administration:** Edward Kroc.

**Resources:** Edward Kroc.

**Software:** Edward Kroc.

**Supervision:** Edward Kroc.

**Validation:** Edward Kroc.

**Visualization:** Edward Kroc.

**Writing – original draft:** Edward Kroc.

**Writing – review & editing:** Edward Kroc.

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
