## [Decision Letter · Decision Letter 0]

13 Jan 2023

PONE-D-22-29796Generalized measurement error: Intrinsic and incidental measurement errorPLOS ONE

Dear Dr. Kroc,

Thank you for submitting your manuscript to PLOS ONE. After careful consideration, we feel that it has merit but does not fully meet PLOS ONE’s publication criteria as it currently stands. Therefore, we invite you to submit a revised version of the manuscript that addresses the points raised during the review process.

The reviewers' comments pointed out that a significant revision is required to improve the current version of the manuscript.  As raised by Reviewer 3, the manuscript must be restructured to meet PLoS ONE's main template (abstract, introduction, materials and methods, results, discussion and conclusion). In addition, the main goal of this work should be better specified and the Introduction should be improved. The mathematical formalism used in the manuscript needs to be carefully revised. A comparison with similar statistical approaches is required. The limitations and possible extensions of the proposed approach need to be clearly defined and explained. Please avoid vague terms and sentences throughout the manuscript. Finally, all the Reviewers highlighted that spell-checking and typo proofreading of the manuscript is required.

We look forward to receiving your revised manuscript.

Kind regards,

Andrea Tangherloni

Academic Editor

PLOS ONE

Journal Requirements:

Reviewers' comments:

Reviewer's Responses to Questions

**Comments to the Author**

1. Is the manuscript technically sound, and do the data support the conclusions?

Reviewer #1: Partly

Reviewer #2: Partly

Reviewer #3: Yes

2. Has the statistical analysis been performed appropriately and rigorously? 

Reviewer #1: No

Reviewer #2: No

Reviewer #3: Yes

3. Have the authors made all data underlying the findings in their manuscript fully available?

Reviewer #1: No

Reviewer #2: No

Reviewer #3: Yes

4. Is the manuscript presented in an intelligible fashion and written in standard English?

Reviewer #1: Yes

Reviewer #2: Yes

Reviewer #3: Yes

5. Review Comments to the Author

Reviewer #1: In this article, there is no novelty, and does not fall in the scope of the journal. ......................................................................................................................................

Reviewer #2: The paper is interesting however it presents several aspects to be improved:

-it needs some spell checking and typo proof-reading. For instance in the abstract there are .. after process and so it happens similarly throughout the manuscript (in the introduction there is a capital letter after semicolon and so on)

- line 10 define the measurable space

-line 13 what is the naive sense?

-define all of the variables/functions used in all of the equations. For instance equation 2 and also other are not defined

These cannot be defined as in equation 3 where there is the use of adjectives this/that but should specified through the use of variables names

-line 25 the goal of the paper is not well specified and is too hidden in the middle of the introduction

-the overall impression of the paper is the one of a report. Many qualitative expressions are reported, as for instance in line 96, so everything should be restructured and rewritten according to a better definition and mathematical formalism of the paper itself

-more applicative examples should be presented for the purpose of evaluation of the proposed measurements error definition

-Figure 3 is not clear, and should be explained better

-How do you compare to similar statistical approaches? A wide comparison should be defined

Reviewer #3: The paper presents a general notion of measuring errors, extending Berkson's work. The findings in this paper are useful and well presented in relationship with the classic Berkson calibration approach. The conclusions support the findings. However, some minor comments were attached in a .pdf document before my final approval.

6. PLOS authors have the option to publish the peer review history of their article (what does this mean?). If published, this will include your full peer review and any attached files.

Reviewer #1: No

Reviewer #2: No

Reviewer #3: No

---

## [Decision Letter · Decision Letter 1]

20 Mar 2023

PONE-D-22-29796R1Generalized measurement error: Intrinsic and incidental measurement errorPLOS ONE

Dear Dr. Kroc,

Thank you for submitting your manuscript to PLOS ONE. After careful consideration, we feel that it has merit but does not fully meet PLOS ONE’s publication criteria as it currently stands. Therefore, we invite you to submit a revised version of the manuscript that addresses the points raised during the review process.

The reviewers' comments highlighted that a detailed discussion of the current state-of-the-art should be done. In addition, the architecture of the proposed model needs to be shown and described in detail. They also pointed out that spell-checking and typo proofreading of the manuscript are required. Please refer to all the reviewers' reports for detailed comments which can help improve the current version of the manuscript.

We look forward to receiving your revised manuscript.

Kind regards,

Andrea Tangherloni

Academic Editor

PLOS ONE

Reviewers' comments:

Reviewer's Responses to Questions

**Comments to the Author**

1. If the authors have adequately addressed your comments raised in a previous round of review and you feel that this manuscript is now acceptable for publication, you may indicate that here to bypass the “Comments to the Author” section, enter your conflict of interest statement in the “Confidential to Editor” section, and submit your "Accept" recommendation.

Reviewer #2: All comments have been addressed

Reviewer #4: (No Response)

Reviewer #5: (No Response)

2. Is the manuscript technically sound, and do the data support the conclusions?

Reviewer #2: Yes

Reviewer #4: Yes

Reviewer #5: Yes

3. Has the statistical analysis been performed appropriately and rigorously? 

Reviewer #2: Yes

Reviewer #4: No

Reviewer #5: Yes

4. Have the authors made all data underlying the findings in their manuscript fully available?

Reviewer #2: No

Reviewer #4: Yes

Reviewer #5: Yes

5. Is the manuscript presented in an intelligible fashion and written in standard English?

Reviewer #2: Yes

Reviewer #4: Yes

Reviewer #5: Yes

6. Review Comments to the Author

Reviewer #2: The manuscript has been thoroughly revised.

I suggest a re-read and improvement of the english form.

Reviewer #4: This article generalizes the traditional notion of measurement error model by allowing the observed data to random valued. The idea that observed data can represent certainty/probability distribution instead of being real valued is interesting. As demonstrated by many examples, this framework could be useful for many applications. Generalized versions of the classical and Berkson error models are developed along with estimation strategies. Some numerical studies are conducted to investigate the performance of the proposed estimators. Given that this article does not contain any real data analysis the scope of numerical studies should be wider. Experiment with different scenarios. Particularly, studies designed to perform a comparison between the new framework and the traditional framework should be conducted.

See attachment.

Reviewer #5: There is a large body of literature surrounding measurement error, its impact, and how to correct for it in the “classical” sense that the author seeks to generalize. However, this author calls this type of measurement error “inessential.” Plenty of others have shown the bias and variance inflation attributable to ignoring this type of measurement error, so I cannot see how it would not be essential to correct for it. I am generally uneasy about the paper’s treatment of the established measurement error literature and feel that it detracted from the overall message.

For readers like myself who are accustomed to “incidental” measurement error, I think further discussion and nuance is needed to first differentiate between the two types being proposed. Some of the points currently used to distinguish between them I do not agree with as written. For example, “intrinsic” measurement error is said be observation-specific, but existing approaches to “incidental” measurement error can allow for this (perhaps to a different degree), too. Without a clear distinction between the types, it can be hard to follow the rest of the paper and see the larger contributions of the proposed generalized framework.

Some other points for the author’s consideration are below.

- If much of the literature focuses on real numbers, is it not because most applications involve real numbers? To understand the importance of this manuscript’s extension to non-real numbers, I think some justification is needed for why this hasn’t been handled in the past but is needed. Also, the first three examples given all seem to have real-valued measurements.

- “Measurement uncertainty is allowed to be potentially unique to each element.” I often see methods and applications in the literature where measurement error is allowed to depend on other factors (e.g., study site or machine type). I don’t see how this (“unique to each element”) differentiates the author’s two types of measurement error.

- In Equation (2), I expected the decomposition to be f(x*,x) = f(x|x*)f(x*), since it would only require modeling f(x|x*) since f(x*) is fully observed.

- In Equation (3), Y and Z have not yet been defined.

- The way the names for the types of measurement error are introduced (e.g., “intrinsic or essential”), to me, imply that the two (“intrinsic” and “essential”) names are synonymous, but they are not.

- Why is the classical measurement error model introduced in Section 1 but Berkson not (since it will be revisited later)?

- In the sentence after Equation (6) there is an i.e., inside of an i.e.,.

- Though I appreciate the many examples, this is a pretty lengthy paper; I think only one or two examples should be kept in the main text and perhaps the rest moved to the supplement.

- R code should be included as an R script (a .R file), rather than printed in a PDF Supplement. It is cumbersome for readers to copy and paste code from a PDF, and also can be prone to errors. The ready-to-run code for the paper should be available online, for example, on figshare or GitHub.

(These are the same comments in my attachment, in case that is easier to read.)

7. PLOS authors have the option to publish the peer review history of their article (what does this mean?). If published, this will include your full peer review and any attached files.

Reviewer #2: No

Reviewer #4: No

Reviewer #5: No

---

## [Decision Letter · Decision Letter 2]

22 May 2023

Generalized measurement error: Intrinsic and incidental measurement error

PONE-D-22-29796R2

Dear Dr. Kroc,

We’re pleased to inform you that your manuscript has been judged scientifically suitable for publication and will be formally accepted for publication once it meets all outstanding technical requirements.

Kind regards,

Andrea Tangherloni

Academic Editor

PLOS ONE

Additional Editor Comments (optional):

Reviewers' comments:

Reviewer's Responses to Questions

**Comments to the Author**

1. If the authors have adequately addressed your comments raised in a previous round of review and you feel that this manuscript is now acceptable for publication, you may indicate that here to bypass the “Comments to the Author” section, enter your conflict of interest statement in the “Confidential to Editor” section, and submit your "Accept" recommendation.

Reviewer #2: All comments have been addressed

2. Is the manuscript technically sound, and do the data support the conclusions?

Reviewer #2: Yes

3. Has the statistical analysis been performed appropriately and rigorously? 

Reviewer #2: Yes

4. Have the authors made all data underlying the findings in their manuscript fully available?

Reviewer #2: No

5. Is the manuscript presented in an intelligible fashion and written in standard English?

Reviewer #2: Yes

6. Review Comments to the Author

Reviewer #2: The manuscript has improved significantly and it now deserves publication

I think it would still benefit from a thorough reread for checking syntax and english form

7. PLOS authors have the option to publish the peer review history of their article (what does this mean?). If published, this will include your full peer review and any attached files.

Reviewer #2: No

---

## [Editor Report · Acceptance letter]

19 Jun 2023

PONE-D-22-29796R2 

Generalized measurement error: Intrinsic and incidental measurement error 

Dear Dr. Kroc:

I'm pleased to inform you that your manuscript has been deemed suitable for publication in PLOS ONE. Congratulations! Your manuscript is now with our production department. 

Kind regards, 

on behalf of

Dr. Andrea Tangherloni 

Academic Editor

PLOS ONE